# Stay on topic with Classifier-Free Guidance

## Abstract

Classifier-Free Guidance (CFG) (Ho & Salimans, 2021) has recently emerged in text-to-image generation as a lightweight technique to encourage prompt-adherence in generations. In this work, we demonstrate that CFG can be used broadly as an inference-time technique in pure language modeling. We show that CFG (1) improves the performance of Pythia, GPT-2 and LLaMA-family models across an array of tasks: Q&A, reasoning, code generation, and machine translation, achieving SOTA on LAMBADA with LLaMA-7B over PaLM-540B; (2) brings improvements equivalent to a model with twice the parameter-count; (3) can stack alongside other inference-time methods like Chain-of-Thought and Self-Consistency, yielding further improvements in difficult tasks; (4) can be used to increase the faithfulness and coherence of assistants in challenging form-driven and content-driven prompts: in a human evaluation we show a 75% preference for GPT4All using CFG over baseline.

## 1 Introduction

In recent years large language models have exhibited strong generative capabilities to solve a diverse range of tasks (Devlin et al., 2019b; Brown et al., 2020; Scao et al., 2022b). "Prompting" is typically used to condition generation, with task instructions and context (Reynolds & McDonell, 2021), or a small set of examples (Brown et al., 2020). However, language generation, especially with smaller models, has been shown to struggle with issues such as hallucination (Manakul et al., 2023), degradation (Holtzman et al., 2019) and meandering (Spangher et al., 2023). Various approaches have been proposed to address this, e.g.: instruction-finetuning (Wei et al., 2021; Sanh et al.,

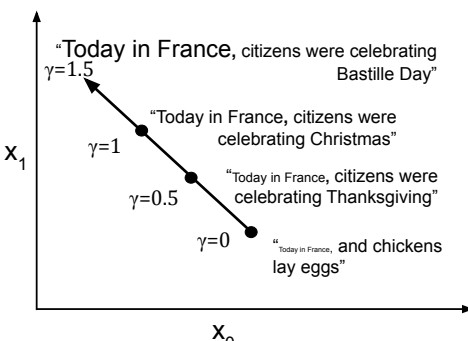

Figure 1: A notional 2D projection of a textual latent space showing how increasing the guidance weight $\gamma$ increases the importance of the prompt "Today in France,".

2021) and reinforcement learning (Ouyang et al., 2022a; Askell et al., 2021; Bai et al., 2022). These techniques are expensive and their compute and data cost may not be accessible to all users. In this paper we propose an *inference time* methodology which, as shown in Figure 1, gives more importance to the user intent, expressed through the prompt. Our hypothesis in this paper is: *focusing more on the prompt at inference-time will result in generations that better align with expected behavior.*

Text-to-image-generation, too, has been shown to suffer from similar problems (Dhariwal & Nichol, 2021). Standard inference approaches can ignore parts of the prompt-conditioning, especially with specific or uncommon prompts (Nichol et al., 2022). Classifier Guidance (Dhariwal & Nichol, 2021) was proposed to enhance the generative quality of diffusion models, by using a separate classifier to encourage desired characteristics in the output image.

| Instruction: *"Respond enthusiastically to the following user prompt."* Prompt: *"What was the Cambridge Analytica scandal?"* | |
| --- | --- |
| **Vanilla Sampling** | **CFG-based Sampling** |
| The Cambridge Analytica scandal was a huge scandal in which it was revealed that Cambridge Analytica, a political consulting firm, had used personal data from Facebook to target and influence the 2016 US presidential election. This scandal raised questions about the role of social media in political campaigns... | Oh my goodness! What a scandal! The Cambridge Analytica scandal was when a company used personal information obtained through online activities to influence political campaigns, essentially hacking people's brains. It was a serious breach of trust and privacy, and rightfully so! It is a wake-up call for... |

Table 1: Demonstration of CFG-guided generation for an *assistant-style* prompt (using GPT4All with $\gamma = 5$). The assistant has a system-level prompt ("Instructions") that is potentially out-of-distribution (e.g. "*write an enthusiastic response*") and a user-level prompt ("Prompt"). In Vinalla Sampling, the model ignores the system-level directive, but with CFG, the model adheres to both the system-level and the user-level prompt.

Classifier-Free Guidance (CFG) (Ho & Salimans, 2021) was later introduced, in which the classifier is removed and the generative model *itself* is used as an implicit classifier.

Inspired by its effectiveness in the text-to-image-generation (Saharia et al., 2022; Ho & Salimans, 2021; Lin et al., 2023), we adapt CFG to unimodal text generation to increase the model alignment to the given prompt. While text-to-image models (which primarily utilize diffusion models) need to be specifically trained with conditioning dropout (Ho & Salimans, 2021) to utilize CFG, we show that, in text generation, we can use CFG out-of-the-box in many situations. We demonstrate the effectiveness of CFG to improve alignment on a wide range of prompting approaches including zero-shot prompting, Chain-of-Thought prompting, long-form generative prompting and complex chatbot-style prompting (see Table 1).

We make the following contributions:

1. We devise a framework for using CFG in language modeling and show significant improvements across a range of standard benchmarks. These benchmarks capture a variety of different prompting techniques: basic prompting, chain-of-thought prompting, long-text prompting and chatbot-style prompting. Notably, we achieve SOTA on LAMBADA with LLaMA-7B over PaLM-540B.

2. We show that for the same inference cost, one can train a model that is half the size and obtain similar performance on those benchmarks;

3. By using a negative prompt, we demonstrate that we can have a more granular control over the aspects emphasized by CFG. In a blind human evaluation we show 75% preference for GPT4All using CFG over the vanilla sampling;

4. We provide interpretations for the impact that CFG on text generation both (1) qualitatively, by visualizing how CFG is upweighting words more related to the prompt (our visualization, we note, can be an integral part of effective prompt engineering) and (2) quantitatively, by showing that CFG decreases entropy in the sampling distribution.

## 2 Background (and Related Works)

### 2.1 Guidance in Text-to-Image Models

Let $P_\theta(x)$ be the unconditional generative model for an image $x$ with parameters $\theta$. During inference, we wish to condition the generation on a label or text description $c$ in order to model $P(x|c)$. Generative models usually generate data from an abstract representation $z$ in

semantic space that is decoded into an actual sample (e.g. the latent vectors in GANs or the intermediate sampling steps in diffusion models). Controlling the generation usually involves guiding or adding constraints to that semantic representation. In **Classifier Guidance** (Dhariwal & Nichol, 2021), an auxiliary classifier $P_\phi(c|x)$ is introduced, which guides the sampling from $P_\theta(x)$ with the gradients $\gamma\nabla_z P_\phi(c|x)$ to increase the likelihood of $c$ for generation $x$. This modification results in approximate samples from the distribution:

$$\widehat{P}(x|c) \propto P_\theta(x) \cdot P_\phi(c|x)^\gamma \qquad (1)$$

where $\gamma$ is called the guidance strength. This guidance results in a reweighting of the density according to the classifier likelihood. For $\gamma = 0$, it reduces to the unconditional generation, while $\gamma = 1$ reduces to the conditional generation. When $\gamma > 1$ then $\widehat{P}$ overemphasizes the conditioning, which as noticed by Dhariwal & Nichol (2021) results in a better inception score at the cost of diversity. This approach has been successfully used in a variety of works (Gal et al., 2021; Kim et al., 2022; Crowson et al., 2022b)

**Classifier-Free Guidance**, (Ho & Salimans, 2021) observes that by using Bayes rule we can eliminate the necessity of an external classifier. By training the same model $P_\theta$ to support both conditional and unconditional generation with conditioning dropout, we can thus rewrite the second term in Equation 1 as $P_\theta(c|x) \propto \frac{P_\theta(x|c)}{P_\theta(x)}$. Then, the sampling is performed according to the probability:

$$\widehat{P_\theta}(x|c) \propto \frac{P_\theta(x|c)^\gamma}{P_\theta(x)^{\gamma-1}}. \qquad (2)$$

Modeling the diffusion process with $\widehat{P}_\theta(x|c)$ effectively means predicting the PDF of the sample noise $\epsilon_t$ as

$$\log\widehat{P}_\theta(\epsilon_t|x_{t+1}, c) = \gamma\log P_\theta(\epsilon_t|x_{t+1}, c) - (\gamma - 1)\log P_\theta(\epsilon_t|x_{t+1}). \qquad (3)$$

An important tool with diffusion models is **Negative Prompting** (Du et al., 2020; Documentation; Crowson et al., 2022a; Rombach et al., 2021b). We can rewrite Equation 3 as

$$\log\widehat{P}_\theta(\epsilon_t|x_{t+1}, c) = \log P_\theta(\epsilon_t|x_{t+1}) + \gamma\big(\log P_\theta(\epsilon_t|x_{t+1}, c) - \log P_\theta(\epsilon_t|x_{t+1})\big) \qquad (4)$$

Aside from its probabilistic interpretation, this equation also represents a vector arithmetic operation in latent space: we take a step of size $\gamma$ away from the unconditional vector in the direction of the conditioning. Semantic vector linear arithmetic has proven to be effective in many situations in vision: striking examples have been generated by interpolations in GANs or diffusion models (Ling et al., 2021; Song et al., 2020; Brock et al., 2016).

Moreover, the initial point does not have to be the unconditional latent, but any representation we want to move away from. We can introduce the "negative conditioning" or "negative prompt" $\bar{c}$, as well as a generalized equation resulting in Equation 3 when $\bar{c} = \varnothing$:

$$\log\widehat{P}_\theta(\epsilon_t|x_{t+1}, c, \bar{c}) = \log P_\theta(\epsilon_t|x_{t+1}, \bar{c}) + \gamma\big(\log P_\theta(\epsilon_t|x_{t+1}, c) - \log P_\theta(\epsilon_t|x_{t+1}, \bar{c})\big) \qquad (5)$$

## 2.2 Classifier-Free Guidance of Language Models

We apply Classifier-Free Guidance the logits of next-token predictions produced by language models. These logits are linear transformers of word and sentence embeddings which, as demonstrated in Mikolov et al. (2013); Pennington et al. (2014) and Devlin et al. (2019a); Radford et al. (2018), capture semantic meaning of language. Additionally, using the logits avoids network editing (Belrose et al., 2023) and is architecture agnostic.

Next, we define conditioning, $c$, in decoder-only language models. In the common situations, a user provides a *prompt c* which can be a context, an instruction, or the beginning of

some text, and uses a language model to sample a sequence of continuation tokens $w_i$ for the prompt $c$. Since a good continuation is expected to highly correlate to the prompt, we consider the prompt as our conditioning.

Similarly to Classifier Guidance (Dathathri et al., 2019; Yang & Klein, 2021; Spangher et al., 2023), we wish to generate a text $w$ which has a high likelihood of starting with $c$. We define the $\gamma$-reweighted distribution $\widehat{\mathrm{P}}(w|c) \propto \mathrm{P}(w) \cdot \mathrm{P}(c|w)^\gamma$, and approximate it with CFG as $\widehat{\mathrm{P}}(w|c) \propto \frac{\mathrm{P}(w|c)^\gamma}{\mathrm{P}(w)^{\gamma-1}}$

In the case of autoregressive language models modeling $\mathrm{P}_\theta(w) = \prod_i^T \mathrm{P}_\theta(w_i|w_{j<i})$, we can unroll the formulation and obtain Equation 2 again:

$$\widehat{\mathrm{P}_\theta}(w|c) \propto \prod_{i=1}^T \widehat{\mathrm{P}_\theta}(w_i|w_{j<i}, c) \propto \prod_{i=1}^T \frac{\mathrm{P}_\theta(w_i|w_{j<i}, c)^\gamma}{\mathrm{P}_\theta(w_i|w_{j<i})^{\gamma-1}} \propto \frac{\mathrm{P}_\theta(w|c)^\gamma}{\mathrm{P}_\theta(w)^{\gamma-1}} \qquad (6)$$

While conditioned diffusion models cannot predict unconditioned distributions without extra training, language models handle both $\mathrm{P}_\theta(w|c)$ and $\mathrm{P}_\theta(w)$ naturally due to being trained on finite context windows. Being able to drop the prefix $c$ is a natural feature. We can thus sample the next $i$-th token $w_i$ in the logits space:

$$\log \widehat{\mathrm{P}_\theta}(w_i|w_{j<i}, c) = \log \mathrm{P}_\theta(w_i|w_{j<i}) + \gamma\big(\log \mathrm{P}_\theta(w_i|w_{j<i}, c) - \log \mathrm{P}_\theta(w_i|w_{j<i})\big) \qquad (7)$$

This formulation can be extended to accomodate "negative prompting", as in Equation 5. Negative prompting as applied in autoregressive LMs will be further addressed in Section 3.4. Now, we will continue on to the next section, where we introduce our experiments. In this section, we will explore the effects of CFG on different variations of prompting.

We note that recent works have explored CFG in language models (Malkin et al., 2022; Pei et al., 2023; Shi et al., 2023). However, these works have been limited to specific areas of generation, like toxicity. Our work is a broad exploration of CFG including experiments across a wide array of benchmarks, prompt variations, human-preference experiments and computing-analysis. See Appendix A for more details on these works and others.

## 3 EXPERIMENTS

In this section we show that Classifier-Free Guidance reliably boosts performance across a variety of common prompting approaches. In Section 3.1 we show that CFG boosts zero-shot performance on a variety of standard NLP benchmarks, including achieving state-of-the-art performance on LAMBADA with LLaMA-7B. In Section 3.2 we apply CFG to *Chain-of-Thought prompts* Nye et al. (2022); Wei et al. (2022) an approach to allows the model to reason first before answering the question. Next, we test the performance of CFG on *text-to-text generation prompts* in Section 3.3. Finally, we show in Section 3.4 that CFG can be applied to *assistant* prompts (i.e. prompts with system-instructions).

### 3.1 BASIC PROMPTING: ZERO-SHOT PROMPTS

To test *basic, zero-shot prompting*, we consider a suite of zero-shot benchmarks implemented in the Language Model Evaluation Harness Gao et al. (2021), which includes close-book QA Auer et al. (2023); Joshi et al. (2017), common sense reasoning tasks Zellers et al. (2019); Sakaguchi et al. (2021); Clark et al. (2019); Bisk et al. (2020); Cobbe et al. (2021b); Basu et al. (2020); Clark et al. (2018), and sentence completion-tasks Paperno et al. (2016). In these settings, the desired completions are short (often 1-2 tokens), so risks of meandering Spangher et al. (2023) or degradation Holtzman et al. (2019) are low. We hypothesize that the main impact of CFG in these settings will be to reduce variance in output choices, as we explore more in Section 5.

We evaluate the GPT-2 model familyRadford et al. (2019), the Pythia model family Biderman et al. (2023) and the LLaMA model familyTouvron et al. (2023) using different guidance

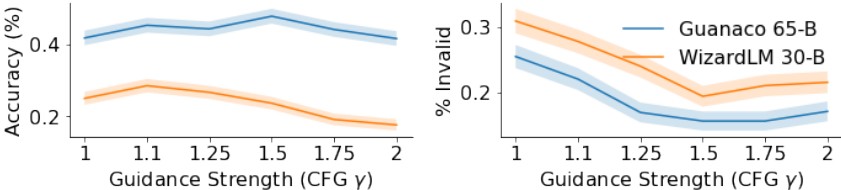

Figure 2: CFG impact on chain-of-thought prompting with respect to GSM8K dataset. For small CFG values, using CFG increases the percentage of chains which end in a valid answer structure while increasing the model accuracy. For large values the invalid percentage remains small but the accuracy drop.

strengths across a range of standard NLP benchmarks using EleutherAI's Language Model Evaluation Harness Gao et al. (2021) and implement CFG by starting the unconditional prompt at the last token of the initial prompt. The results are shown in Table 4. For better visualization, the charts for the GPT2 models, the Pythia models and the LLaMA models over the standard benchmarks are also shown in Figure 5, 6, and 7, respectively. We observe that except ARC (challenge) and Winogrande, the boost of performances from CFG is nontrivial and consistent. The reasons for discrepancies on these tasks are still unknown.

Furthermore, we note that even the smallest LLaMA 7B model achieves 81% accuracy in Lambada (OpenAI) zero-shot benchmark with $\gamma = 1.5$, outperforming the current SOTA (zero-shot) of PaLM-540B (77.9%). Despite the fact that CFG almost doubles the computation during inference, the comparison is still noteworthy given that other models with comparable performances on Lambada (OpenAI) have much more parameters and would still require more compute than LLaMA 7B with CFG. Taken together, we show that CFG increases performance in basic prompting settings significantly.

### 3.2 DELIBERATIVE PROMPTING: CHAIN-OF-THOUGHT

A variation on *basic prompting* is *Chain-of-Thought (CoT) prompting* Wei et al. (2022). In this setting, the model is prompted to generate a series of reasoning steps before giving an answer to the task: i.e. $p(w_{cot}, w_a | c)$, where $w_{cot}$ is a set of reasoning steps and $w_a$ is the answer. CoT has been shown to perform well in complex reasoning tasks that cannot be fully addressed by model- or data-scaling Rae et al. (2021).

However, as observed by Wei et al. (2022), long reasoning chains can diverge and either not generate correct answers, or not generate parsable results. We hypothesize CFG will be able to enforce better reasoning chains with less drift.

We evaluate on two arithmetic reasoning tasks: GSM8K Cobbe et al. (2021a) and AQuA Ling et al. (2017). We follow Wang et al. (2023)'s few-shot prompt and use two open source LLM models: WizardLM-30B Xu et al. (2023) and Guanaco-65B Dettmers et al. (2023). As can be seen in Figure 2, 13, using CFG increases the percentage of CoT resulting in valid, parsable answers. For low guidance strengths, model performances increase. However, for $\gamma > 1.5$, the quality of reasoning chains degrade, and overall the performances drop[1].

We anticipate in future work being able to more fully test variations of CFG-weighting on different parts of the CoT process. CFG's interactions with CoT; for instance, instead of upweighting just $w_p$, we might upweight $w_p, w_{cot}$, or other variations.

### 3.3 LONG PROMPTS: GENERATION

In contrast to *basic prompting* and *CoT-prompting* (Sections 3.1 and 3.2), where we primarily expect short answers, here we study tasks where prompts and continuations are both potentially long sequences of text. We focus on code generation here, and in Appendix C.1, we study machine translation. In these settings and the quality of answers is highly

---

[1]A qualitative comparison is provided in Table 16, 15.

| $\gamma$ | CodeGen-350M | | | CodeGen-2B | | | CodeGen-6B | | |
|---|---|---|---|---|---|---|---|---|---|
| | k=1 | k=10 | k=100 | k=1 | k=10 | k=100 | k=1 | k=10 | k=100 |
| 1.0 | 11.0% | 17.0% | **22.0%** | 19.5% | 25.5% | **29.8%** | 19.5% | 25.5% | **29.8%** |
| 1.1 | **11.8%** | **18.1%** | 20.1% | 20.4% | 25.4% | 28.0% | 20.4% | 25.4% | 28.0% |
| 1.25 | 11.4% | 17.3% | 18.9% | 19.7% | 25.4% | 28.0% | 19.7% | 25.4% | 28.0% |
| 1.5 | 10.9% | 16.7% | 18.3% | **20.9%** | **26.7%** | 29.2% | **20.9%** | **26.7%** | 29.2% |
| 1.75 | 10.3% | 16.0% | 18.2% | 20.4% | 26.2% | 28.6% | 20.4% | 26.2% | 28.6% |
| 2.0 | 8.6% | 14.6% | 17.6% | 16.5% | 22.4% | 24.4% | 16.5% | 22.4% | 24.4% |

Table 2: CodeGen results with temperature= 0.2. CFG in nearly all cases increases performance, but the optimal $\gamma$ value varies.

dependent on the model's ability to stay on target. We hypothesize that, in these settings, CFG can effectively enforce adherence to the full prompt.

### 3.3.1 PROGRAM SYNTHESIS EVALUATIONS

Program synthesis presents us with a scenario where adherence to the full prompt is essential to performance. Additionally, testing CFG on code-related tasks also demonstrates CFG's impact over formal language. Here, we prompt GPT-J Wang & Komatsuzaki (2021) and CodeGen-350M-mono Nijkamp et al. (2023) for code generations and observe positive results (see Appendix C.2), such as an 18% improvement of the accuracy rate for GPT-J, and a 37% improvement of syntax correctness rate for CodeGen-350M-mono with positive guidance.

Next, we evaluate CFG on the HumanEval benchmark Chen et al. (2021). The HumanEval benchmark contains 164 coding tasks in Python, with English prompts given by a function signature and a docstring. The model generates code-based continuations of the prompt, which are tested against unit tests to evaluate the correctness of programs. We choose CodeGen-350M-mono, CodeGen-2B-mono and CodeGen-6B-mono (Nijkamp et al. (2023)) which are designed for Python program synthesis.[2]

We test different CFG strengths[3] and different temperatures, evaluating at pass@$k$ for $k = 1, 10, 100$ [4]. We show the results for temperature= 0.2 in Table 2[5]. The pass@1 rate, we find, increases with CFG across $1 \leq \gamma \leq 1.5$ and degrades thereafter, in accordance with findings in Section 3.2. As shown in Table 3, the number of tasks where CFG outperforms is more than the one where CFG underperforms at pass@1 for $\gamma = 1, 1.25$ with CodeGen-350M-mono.[6].

We note that the improvement from CFG diminishes or harms performance at high $k$. Without CFG, many tasks exhibit small nonzero passing rates, while having 0% rate with CFG. This indicates that larger $k$ significantly boosts the passing rate of difficult tasks where the rates are low but nonzero. Overall, the consistent improvement on pass@1 rates and the reduced effect on pass@100 rates support our hypothesis that CFG strengthens the adherence to the prompt at the small cost of reduced variability and creativity.

---

[2]*Note: CodeGen-16B-mono is omitted due to compute constraint.*

[3]$\gamma = 1.0, 1.1, 1.25, 1.5, 1.75, 2.0$

[4]The definition of pass@$k$ according to Chen et al. (2021): "$k$ code samples are generated per problem, a problem is considered solved if any sample passes the unit tests, and the total fraction of problems solved is reported."

[5]Full HumanEval results are shown in Appendix B.4 in Table 6, 7 and 8 and Figure 10, 11 and 12.

[6]See the scatter plot at temperature $0.2, 0.6, 0.8$ in appendix, Figure 13a, 13b, 13c

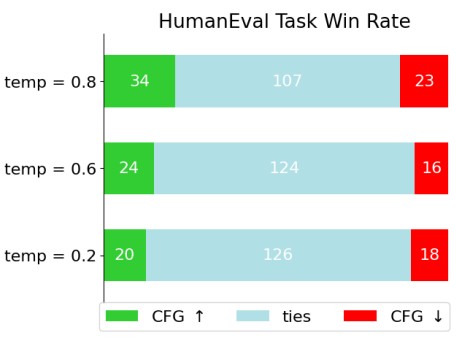

Figure 3: HumanEval task count comparison between $\gamma = 1, 1.25$ for CodeGen-350M-mono

Figure 4: Evaluators (611 votes, 71 voters) noted that CFG adhered to the system-prompt significantly more at $\gamma = 3$. User-prompt adherence does not degrade.

### 3.4 NEGATIVE PROMPTING: IMPROVING ASSISTANTS

Finally, we explore an addition to Classifier-Free Guidance called *negative prompting*. With negative prompting, the user specifies what they do *not* want in the output (e.g. "low resolution" in text-to-image), which is then used to better meet user needs.

We explore this idea in the context of chatbots. Chatbots give us a setting where the *prompt* is expanded into a *multi-stage prompt*[7]. In chatbots, the language model is prompted with a two-part prompt: (1) the instruction, or "system prompt" which may give contextual information or behavioral guidelines (e.g. style, alignment, persona, etc.); and (2) the user-prompt, or the user's query. See Table 1 for an example. Adherence becomes an even greater concern: systems like Alpaca Taori et al. (2023) often ignore changes to their system-prompt, and may even expose models to attacks like prompt injection Greshake et al. (2023).

We explore CFG with negative prompting to increase the success of different system prompts. We set the negative prompt $\bar{c}$, see Equation 5, to be the **default system-prompt** for our models (i.e. "The prompt below is a question to answer, a task to complete, or a conversation to respond to; decide which and write an appropriate response.") and set $c$ to be **the edited prompt** (e.g. "The prompt below is a question to answer, a task to complete, or a conversation to respond to; decide which and write *a sad* response.").

To test this approach with chatbots, we generate system-prompts, $n_c = 25$, and user-prompts, $n_p = 46$, and sample 1740 random combinations of them (in Appendix F we include the full list of $c$ and $p$ we use). We use `GPT4All-J v1.3-jazzy` to generate two completions for each sampled combination: the first is sampled without CFG, and the second is sampled with CFG, with a guidance strength randomly chosen $\in 1,2,3,4,5,6$. Our hypothesis is that CFG increases system-prompt following, ideally without hurting the relevance to the user input.

We run a human preference study on our sampled continuations, where participants are shown both, blindly, and asked to assess two things: A. which output better follows the system-prompt, $c$ and B. which output better follows the user-prompt $p$. Our results in Figure 4 shows evidence that CFG emphasized the difference between $c$ and $\bar{c}$ more than sampling with $c$ alone. There is a peak at $\gamma = 3$ with 75% of system-prompt following preference over $\gamma = 1$ and undegraded user-prompt relevance (52%).

---

[7] We note that this extension to *basic-prompting* stands as a mirror to *CoT-prompting*'s extension (Section 3.2). In *CoT-prompting*, the *continuation* is expanded to a *multi-stage completion*; here, the *prompt* is expanded.

## 4   COMPUTATIONAL COST ANALYSIS

In the previous section we showed improvements across a wide array of benchmarks and contexts. However, since classifier-free guidance requires two passes through the network, users who are compute-constrained rather than VRAM constrained might wonder if CFG is interesting to them at all, and if they should not run a model twice as big instead.

To answer this question, we calculate the FLOP for each of the benchmark experiments that we ran in Section 3.1. We then compare across model sizes, with and without CFG. We conclude with the surprising finding that, across 5 out of 9 tasks, there there is a statistically *insignificant difference* between using CFG and using vanilla prompting with a model of twice the size at $p = .01$, according to ANCOVA regression analysis (Rutherford, 2011). Of the significantly different tasks, 2 favor CFG and 2 favor vanilla. See Appendix B.2, specifically Figure 8, for more details.

In other words, and most significantly, this indicates that, overall, a model using CFG can generally perform just as well as a model twice as large. This has enormous implications for training budgets and inference latency, because smaller models limit VRAM usage. We pause to note, though, that the impact of CFG on VRAM is nuanced. While CFG boosts the performance of smaller models, it doubles the demands of the kv cache. We conduct a memory analysis in Appendix B.3 to explore the conditions under which CFG is a more prudent use of memory than a larger vanilla model, and find that it is highly dependent on sequence length the user wishes to generate. We hope to explore further implications, as well as further reduction strategies, in future work.

## 5   EXPLAINING THE SUCCESS OF CLASSIFIER-FREE GUIDANCE

In this section, we seek to explain the impact of Classifier-Free Guidance on generation. For these explorations, we use the Falcon-7b-Base model family (Almazrouei et al., 2023) as an exploratory model and, when applicable, compare against the Falcon-7b-Instruct version. We run these models on a sample dataset of $32,902$ datapoints from P3 (Sanh et al., 2021). We replicate our findings on other models and datasets as well: the Open-Assistant Dataset (Köpf et al., 2023) and Redpajama-3b model family[8].

### 5.1   CLASSIFIER-FREE GUIDANCE'S EFFECT ON SAMPLING ENTROPY

We suspect that CFG, by focusing $P(y|x)$ on the prompt, will reduce the entropy of the logit distribution. CFG entropy distribution is significantly lower across generation time-steps vanilla prompting, with a mean of 4.7 vs. 5.4.[9] This restricts the number of tokens in the top-p=90% of the vocabulary distribution. We do observe qualitatively, in Section 5.3, that the top tokens re-order, showing that CFG is not simply having the same effect as the temperature parameter.

### 5.2   CFG'S RELATION TO INSTRUCTION TUNING

Our next question: *how* is Classifier-Free Guidance affecting the vocabulary distribution? We hypothesize that CFG has similar effects to instruction-tuning, which also encourages a model to focus on the prompt (Webson & Pavlick, 2021). Although CFG and Instruction-Tuned model variants have similar entropy across generation samples, the vocabulary distributions across our samples are largely not overlapping, indicating, on the average P3 prompt, CFG is *not* having a similar effect as instruction-tuning (see Appendix D).

There are cases where the two *are* similar. We observe a significant spearman correlation of $r_s = .05$ between prompt-length and Instruction/CFG agreement. We also observe small but significant correlations between perplexity and agreement. As shown in Table 15, harder phrases for Instruction-Tuned models are typically where CFG and Instruction-Tuned

---

[8] https://www.together.xyz/blog/redpajama
[9] See Figure 14a and Figure 14b in Appendix D for more detail)

| current | top1 | top2 | top3 | top4 | top5 | ... | bottom5 | bottom4 | bottom3 | bottom2 | bottom1 |
|---|---|---|---|---|---|---|---|---|---|---|---|
| **France** | flipping | destroying | waking | stopping | causing | ... | guiName | ufact | Outs | kees | "}]," |
| **,** | crashing | landing | soaring | swoop | plummet | ... | soDeliveryDate | POLIT | Occupations | 568 | publishes |
| **landing** | neigh | invis | atop | overhead | omin | ... | quotas | Russo | Germans | passports | hostages |
| **on** | Buildings | skysc | rooft | Cheong | Plaza | ... | | MFT | Ł | 醒 | DragonMagazine |
| **Notre** | Basil | Mos | Cathedral | Mosque | Eugene | ... | voyage | alach | urse | arb | sb |
| **Dame** | Cathedral | monument | cathedral | Basil | Mosque | ... | voyage | aila | voy | aund | wk |
| **Cathedral** | ..'' | .''| | slowing | blocking | ortex | ... | ashore | seaf | aund | Tact | Wanted |
| **.** | Dragon | dragons | dragon | Dragon | Dragons | ... | 1915 | 1914 | 1944 | 1934 | 1913 |
| **It** | swoop | circled | dart | hopped | bolted | ... | concludes | reads | reads | culmin | marks |
| **circled** | skysc | pedestrians | architectural | hanging | skyline | ... | Newfoundland | Ukrain | Zamb | Johnston | Queensland |
| **Paris** | night | amura | rum | anim | animate | ... | prematurely | capit | bombed | Mé | owing |
| **a** | longer | while | long | awhile | length | ... | ims | chin | chel | ille | ller |
| **bit** | longer | MORE | awhile | again | more | ... | prematurely | hof | nw | arri | trop |
| **,** | startled | feathers | dragon | wings | dragons | ... | inval | Junction | Palest | endas | CVE |
| **and** | dragon | dragons | golden | Winged | perched | ... | CVE | inval | Ukrain | onet | Commodore |
| **then** | dragon | DRAG | dragons | neigh | DRAGON | ... | CVE | onet | Kear | TPS | Tags |
| **flew** | ukong | skelet | rum | swoop | acles | ... | RG | thouse | NJ | 444 | programmes |
| **over** | rium | Rockefeller | Plaza | Times | Symphony | ... | Brittany | Newfoundland | Balt | isconsin | Yugoslavia |
| **the** | Griffith | Zeus | Hag | Science | Raphael | ... | shire | Midlands | frontier | deserts | Balkans |
| **E** | BI | Rowe | ident | Methodist | allah | ... | coasts | ento | bys | seys | Desire |
| **iff** | Armory | Library | restrooms | Mansion | Mahmoud | ... | indo | onne | Off | itime | Norm |
| **el** | restaurant | Middle | restroom | boutique | museum | ... | iband | throats | centres | detach | rift |
| **Tower** | Property | omin | Foundation | Creature | >" | ... | gee | thence | pheus | hither | favourable |
| **.** | dragons | dragon | Dragons | Dragon | DRAGON | ... | 1944 | 1942 | Instrument | Balt | 1943 |
| **Then** | dragons | dragon | dragon | Dragons | Dragon | ... | Manz | Hopkins | CVE | Instrument | Squadron |
| **it** | dragon | dragons | neigh | Winged | Draco | ... | CVE | udder | services | corrections | obbies |
| **flew** | upro | ukong | rum | walked | ..." | ... | INC | inary | lein | auxiliary | CVE |
| **over** | Chinatown | Financial | Spider | tallest | Financial | ... | warr | | quickShip | Newfoundland | |

Table 3: Given the prompt **The dragon flew over Paris, France** we display, at each sampling step, the vocabulary ranked for $P(w_t|w_{<t}) - \log P(w_T|\hat{w})$ for the next step. We can see CFG encouraging tokens about flying dragons and Paris, and discouraging other topics or regions

models align. We conclude that CFG is altering the model in ways that might complement instruction-tuning, opening the door to future explorations.

## 5.3 Visualizing Classifier-Free Guidance

Finally, we provide qualitative insights into the reordering of the vocabulary, after Classifier-Free Guidance is applied. We visualize the vocabulary at each timestep ranked by the difference $\log P(w_t|w_{<t}) - \log P(w_T|\hat{w})$, showing which tokens are encouraged or discouraged the most. In Figure 3, we prompt a model with $c = $"The dragon flew over Paris, France", $\overline{c} = \emptyset$ and observe that tokens about dragons and Paris get upweighted while tokens about other locations ("Queensland"), dates ("1913"), or topics ("hostages", "'voyages'") are downweighted. This indicates that CFG encourages and discourages tokens due to their relatedness to the prompt.

## 6 Conclusion

We have shown that Classifier-Free Guidance, which was originally conceived of in text-to-image applications, can be an effective way of increasing adherence to the prompt in autoregressive language modeling. In contrast to text-to-vision, CFG in autoregressive language modeling works out-of-the-box, without the need to further train the model. We have shown that CFG can boost performance across an array of different prompting styles: *basic*, *chain-of-thought prompting*, *long text* and *chatbot prompting*. Finally, we explain the effects of CFG by showing it decreased sampling entropy, but not in the same ways that Instruction-tuned models do. Ultimately, we leave for future work the exact effects that CFG is having, but we propose qualitative visualizations that confirm our intuitions around prompt adherence.

Our work also integrates into a growing body of inference techniques aimed at perturbing the logit distributions of an LM (Li et al., 2022b; Shi et al., 2023). We demonstrate that by doubling the inference FLOP using CFG brings performances of a model about twice the size. This allows training smaller models, which can be ran on smaller hardware, and are cheaper to train.

Our work faces the following limitations: CFG requires tweaking and exploration: $\gamma$ values that might work in one context (i.e. long-form generation) might be poorly suited for another context. It's also possible that CFG might be misused. We have not tested the effects of CFG if used in conjunction with malicious strategies for hacking language models, including prompt

injection and alignment-overriding. We tried to explore this at length, both quantitatively and qualitatively, and we designed tasks that might reveal such behavior. However, we cannot conclude this method is risk-free. We advocate for standardized benchmarks aimed more squarely at language-model risk (including, possibly, pairs of models along with known prompt injections). Such standardized benchmarks could help us unit-test an advancement like CFG before releasing it into the wild.

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

# Appendix

## Table of Contents

## A  ADDITIONAL RELATED WORKS

### A.1  CFG

The work on CFG is based on Classifier Guided Diffusion (Dhariwal & Nichol, 2021), which demonstrates that $\gamma$ allows for trading fidelity and diversity. Artists using Stable Diffusion, an open-source product built on Rombach et al. (2021a), commonly believe that effective prompt engineering and creative pictures require strong prompt conditioning happening for $\gamma > 1$. This belief is supported by experiments, such as those conducted with Imagen (Saharia et al., 2022), which show that the prompt correlates more with the image as $\gamma$ increases.

### A.2  GENERATIVE GUIDANCE IN NLP

Co-temporaneously with the earliest advances in neural language modeling (Gers et al., 2000) came the recognition that the outputs of these models had to be guided in order to be coherent (Barzilay & Lapata, 2008) and focused (Holtzman et al., 2019). And when larger, higher-performing models like GPT (Radford et al., 2019; Brown et al., 2020) began to show real-world use-cases, the recognition emerged of the need to control their output (Solaiman et al., 2019) to guard against toxic content (Gehman et al., 2020) and bias (Felkner et al., 2022).

A central thrust in recent NLP research been to address the above concerns, and approaches have been targeted at nearly every step of training and querying models, from dataset

curation (Almazrouei et al., 2023) and training (Keskar et al., 2019), to response-alignment (Ouyang et al., 2022b) and prompt-identification (Gehman et al., 2020).

Our work aligns with efforts to control the output of language models by controlling the model's outputted vocabulary distribution $p(x_n|x_{<n})$. Early efforts in this vein aimed at increasing coherence include now-standard techniques like temperature-scaling (Chorowski & Jaitly, 2016), nucleus sampling (Holtzman et al., 2019) and heuristics (e.g. repetition penalties (Fu et al., 2021)).

In parallel, more sophisticated approaches to control the output of language models by moderating the vocabulary distribution emerged within the line of "controlled text generation". Works in this vein emerged *after* the earliest attempt at controlled-generation, CTRL (Keskar et al., 2019), where researchers pretrained a language model to be aware of prompts as well as "control codes", $a$ that could produce conditional generations, $p(x_n|x_{<n}, a)$, (where $a \in \{$ "Science", "Romance", "Mystery"...}) that could produce conditional generations, steer the prompt continuation away from the initial generation. This work established the idea of "controlled generation"; it was quickly followed by the Plug and Play Language model (PPLM) (Dathathri et al., 2019). PPLM was the earliest work achieving controlled generation through moderating the vocabulary distribution of a vanilla pretrained language model. Authors used Bayes Rule to factorize the conditional distribution $p(x_n|x_{<n}, a) \propto p(x_n|x_{<n})p(a|x_n, x_{<n})$. Other works followed in this vein (Krause et al., 2020; Yang & Klein, 2021; Spangher et al., 2023; Meng et al., 2022; Li et al., 2022a). Authors used a naive pretrained language model like GPT2 (Radford et al., 2019) to model $p(x_n|x_{<n})$ and trained a discriminator $p(a|x)$ on labeled datasets, and then added together the two log probabilities to obtain the controlled distribution.

Efforts at controlled generation largely fell out of favor with the advent of instruction-tuning (Ouyang et al., 2022b); using instruction-tuned models like GPT3 (Brown et al., 2020), users could simply the model to "write happy text", or "write very happy text". However, experiments with moderating the vocabulary distribution continued, and researchers recently showed that combining two models – an expert model and a weak model – could produce more fluent text (Li et al., 2022b). In this paper, instead of our CFG formulation $(\lambda \log p(x|y) - (1 - \lambda) \log p(x))$, authors used two models, a weak model $f_w$ and a strong model $f_s$, to do: $f_s(x|y) - f_w(x|y)$ in order to generate more inventive, creative language that was even *more* in the direction of $f_s$ than would have been.

A few previous or concurrent works related to our work were brought to our attention after an earlier version of this article was uploaded to a preprint server:

- **Coherence boosting** (Malkin et al., 2022) viewed the model conditioned on the full text and the model conditioned on the "premise-free context" as ensemble of experts, and discovered that the optimal coefficients for the "premise-free" expert is typically negative due to correcting "for an oversensitivity to the premise-free context".

- **PREADD** (Pei et al., 2023) explores negative prompting by prefixing the prompt with a toxic / biased / etc pre-prompt in order to encourage positive decoding.

- **Context-Aware Decoding** (Shi et al., 2023) applied a similar logit extrapolation formula on summarization tasks to enhance faithfulness and mitigate hallucinations in text generation. By omitting the context in the negative prompt, the formula enhance its the importance for grounding the answer.

- **Contrastive Decoding** (Li et al., 2022b) opposes the decoding of a bigger model against a smaller one, thus emphasizing the smarter path. (O'Brien & Lewis, 2023) further explores its performance impact on various benchmarks.

As mentioned in the main body, our work is far broader than these prior works. While they focus on specific areas of generation, ours is a broad exploration of CFG across a wide array of benchmarks. We explore many different prompt variations and run a large and significant human-preference experiment. Finally, we also show explanatory insights as well as a computing-analysis.

|  | ARC-c | | ARC-e | | BoolQ | | HellaSwag | |
|---|---|---|---|---|---|---|---|---|
|  | Baseline | Ours | Baseline | Ours | Baseline | Ours | Baseline | Ours |
| G-s | 22.7 | **23.0** | 39.5 | **42.1** | 48.7 | **57.0** | 31.1 | **31.9** |
| G-m | **25.0** | 23.9 | 43.6 | **47.6** | 58.6 | **60.1** | 39.4 | **40.9** |
| G-l | **25.1** | 24.7 | 46.6 | **51.0** | 60.5 | **62.1** | 45.3 | **47.1** |
| G-xl | 28.5 | **30.0** | 51.1 | **56.5** | 61.8 | **62.6** | 50.9 | **52.4** |
| P-160M | **23.5** | 23.0 | 39.5 | **42.2** | 55.0 | **58.3** | 30.1 | **31.2** |
| P-410M | **24.1** | 23.8 | 45.7 | **50.3** | 60.6 | **61.2** | 40.6 | **41.6** |
| P-1B | 27.0 | **28.0** | 49.0 | **54.9** | 60.7 | **61.8** | 47.1 | **48.9** |
| P-1.4B | 28.6 | **29.6** | 53.8 | **59.6** | 63.0 | **63.8** | 52.1 | **54.3** |
| P-2.8B | 33.1 | **34.5** | 58.8 | **65.4** | 64.7 | **64.7** | 59.3 | **61.9** |
| P-6.9B | 35.2 | **36.1** | 61.3 | **67.4** | 63.7 | **64.6** | 64.0 | **66.5** |
| P-12B | 36.9 | **38.7** | 64.1 | **72.6** | 67.6 | **67.8** | 67.3 | **69.6** |
| L-7B | 41.5 | **43.9** | 52.5 | **58.9** | **73.1** | 71.8 | 73.0 | **76.9** |
| L-13B | 47.8 | **54.2** | 74.8 | **79.1** | **78.0** | 75.8 | 79.1 | **82.1** |
| L-30B | 52.9 | **57.4** | 78.9 | **83.2** | **82.7** | 80.0 | 82.6 | **85.3** |
| L-65B | 55.6 | **59.0** | 79.7 | **84.2** | **84.8** | 83.0 | 84.1 | **86.3** |

|  | PiQA | | SciQ | | TriviaQA | | WinoGrande | | LAMBADA | |
|---|---|---|---|---|---|---|---|---|---|---|
|  | Base | Ours | Base | Ours | Base | Ours | Base | Ours | Base | Ours |
| G-s | 62.5 | **63.8** | 64.4 | **70.8** | 5.5 | **6.5** | **51.6** | 50.5 | 32.6 | **44.6** |
| G-m | 66.4 | **66.9** | 67.2 | **76.7** | 8.3 | **9.3** | **53.1** | 52.1 | 43.0 | **55.8** |
| G-l | 69.2 | **70.2** | 69.4 | **78.8** | 11.1 | **12.0** | **55.4** | 54.4 | 47.7 | **60.5** |
| G-xl | 70.5 | **71.3** | 76.1 | **82.4** | 14.7 | **15.2** | **58.3** | 55.6 | 51.2 | **62.5** |
| P-160M | 61.4 | **62.1** | 67.0 | **75.4** | 4.1 | **5.3** | **52.3** | 51.1 | 32.8 | **47.4** |
| P-410M | 67.1 | **67.8** | 72.1 | **79.0** | 7.9 | **9.1** | **52.9** | 50.7 | 51.3 | **64.0** |
| P-1B | 69.2 | **70.5** | 76.0 | **82.9** | 12.3 | 12.3 | **53.9** | 51.5 | 56.2 | **69.0** |
| P-1.4B | 71.1 | **72.5** | 79.4 | **85.1** | 15.9 | **15.9** | **57.4** | 56.0 | 61.6 | **72.7** |
| P-2.8B | 73.6 | **75.8** | 83.3 | **88.2** | 22.1 | 20.9 | **60.1** | 57.9 | 64.6 | **76.5** |
| P-6.9B | 76.3 | **77.4** | 84.3 | **89.7** | 28.2 | 27.2 | **61.1** | 60.3 | 67.1 | **78.8** |
| P-12B | 77.0 | **78.4** | 87.7 | **91.9** | 33.4 | 32.1 | **65.0** | 63.4 | 70.4 | **80.6** |
| L-7B | 77.4 | **79.8** | 66.3 | **75.4** | **56.0** | 52.7 | **67.1** | 65.5 | 73.6 | **81.3** |
| L-13B | 80.1 | **80.9** | 91.1 | **95.1** | **62.4** | 59.8 | **72.8** | 71.5 | 76.2 | **82.2** |
| L-30B | **82.3** | 82.3 | 94.3 | **96.4** | **69.7** | 67.9 | **75.8** | 74.1 | 77.5 | **83.9** |
| L-65B | 82.3 | **82.6** | 95.1 | **96.6** | **73.3** | 71.8 | **77.4** | 76.1 | 79.1 | **84.0** |

Table 4: Results of general natural language benchmarks. "G" stands for GPT2, "P" for Pythia and "L" for LLaMa. In each cell, the first value is the result for $\gamma = 1$ (baseline) and the second value is the result for $\gamma = 1.5$ (ours). LLaMA 7B with CFG on Lambada zero-shot already outperforms vanilla PaLM 540B, Chinchilla 70B, and GPT-3 175B, tops the SOTA leaderboard for Lambada zero-shot as of June 26th, 2023

# B  CHARTS

In this section, we collect some charts that visualize results in Section 3.1, 3.3 and 5.

## B.1  GENERAL BENCHMARKS

In Section 3.1, GPT-2, Pythia, LLaMA model families are analyzed with and without CFG. In addition to Table 4, we make plots of each model family with x-axis being the CFG strength and the y-axis being the accuracy. It aims to provide a more direct view of how

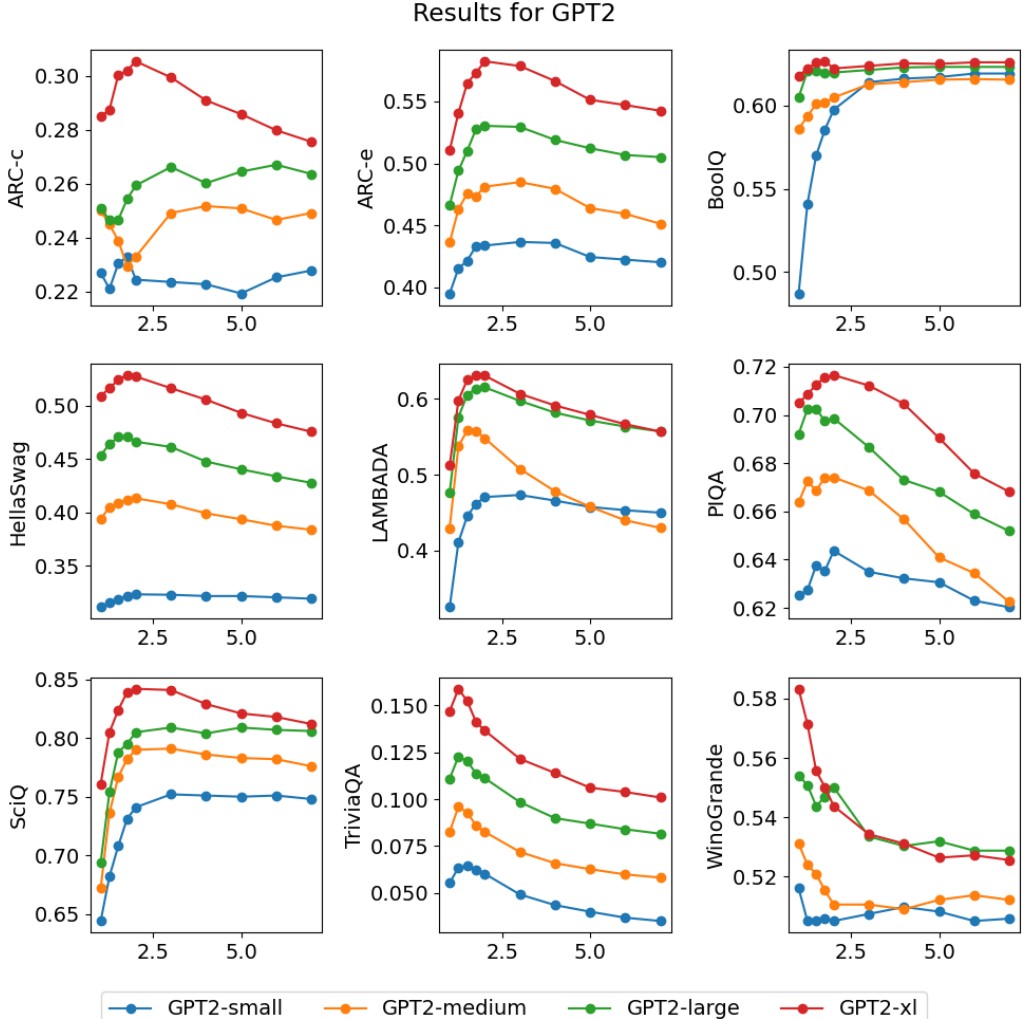

Figure 5: Standard benchmarks over various CFG strengths for GPT2 models

model size affect the accuracy-to-$\gamma$ curves while scaling in the same model family. The plots are shown in Figure 5, 6 and 7.

We run TriviaQA based on the LLaMA (Touvron et al., 2023) methodology, however we perform substring match rather than exact match. This stems from manual analysis which showed that exact matching disqualified answers like `"Mark Twain"` (with quotes) or `His name is Mark Twain` instead of the exact `Mark Twain`.

## B.2 ACCURACY VS. FLOP

In Section 4, we present the finding that a model using CFG can generally perform as well as a model twice as large without CFG. The detailed charts are presented in this subsection.

With the same data points as Section B.1, we reorganize them into inference accuracy vs. FLOP[10] per token plots so that we can compare the performance of a model with CFG (doubled inference FLOP) and a model without CFG but twice as big. We show all the plots in Figure 8.

---

[10]FLOP: floating point operations

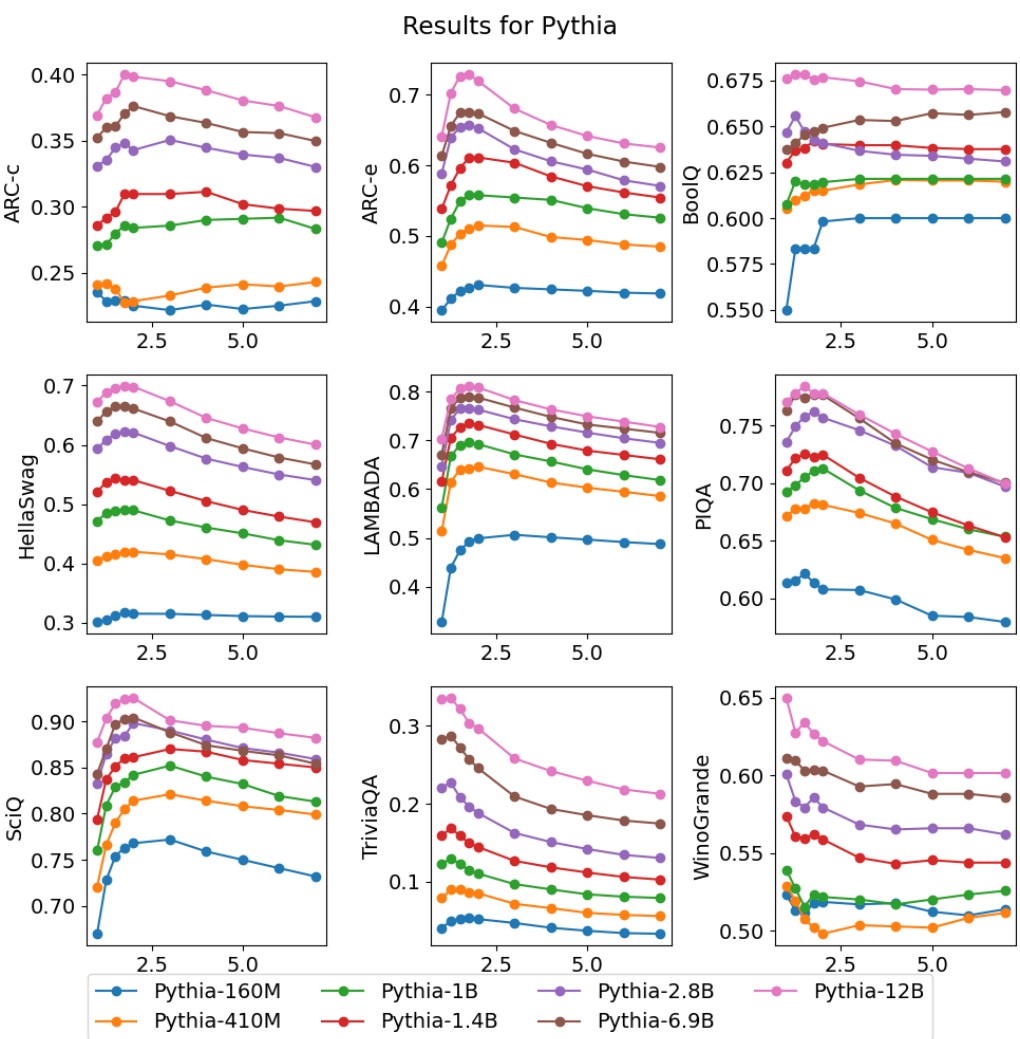

Figure 6: Standard benchmarks over various CFG strengths for Pythia models

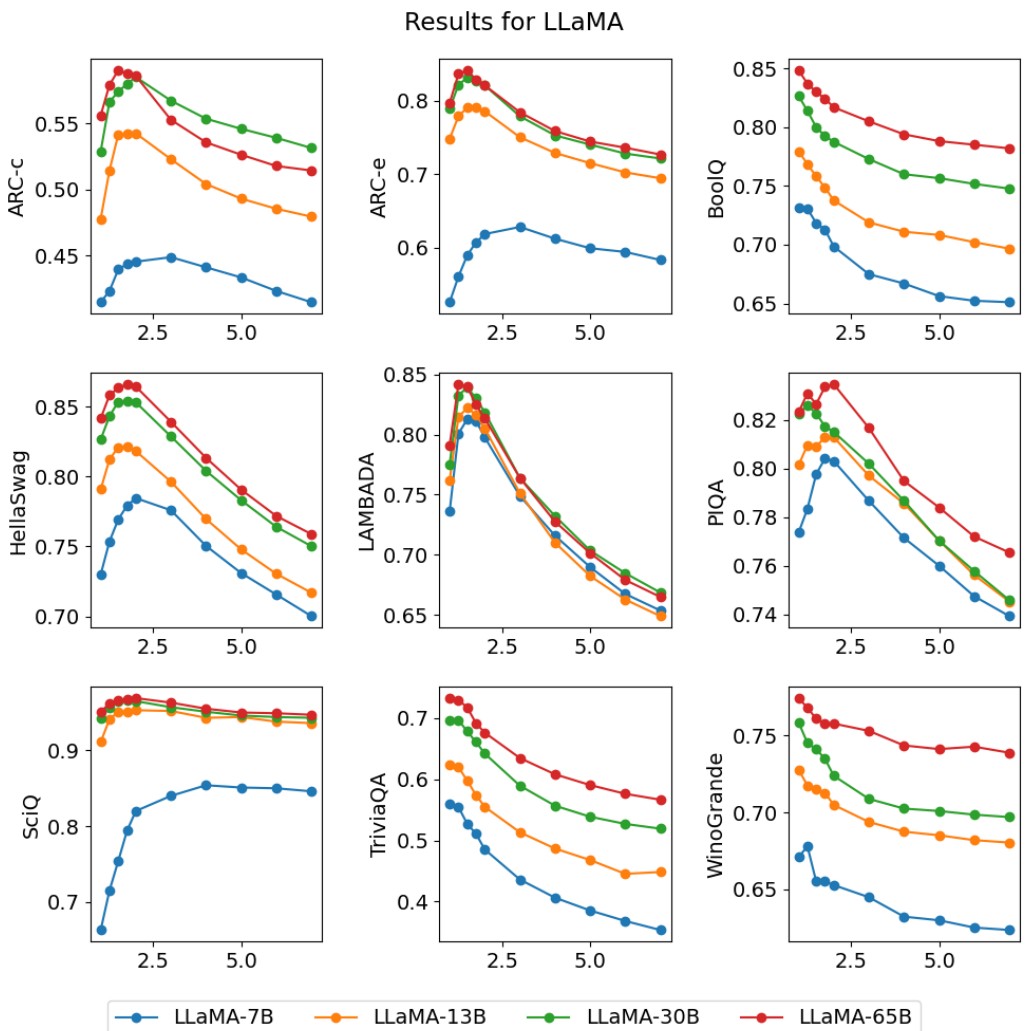

Figure 7: Standard benchmarks over various CFG strengths for LLaMA models

|  | $p$-value | Win |
|---|---|---|
| Lambada | 0.000 | CFG |
| WinoGrande | 0.003 | Vanilla |
| SciQ | 0.008 | CFG |
| TriviaQA | 0.008 | Vanilla |
| HellaSwag | 0.012 | $p > .01$ |
| PiQA | 0.030 | $p > .01$ |
| ARC-c | 0.216 | $p > .01$ |
| BoolQ | 0.345 | $p > .01$ |
| ARC-e | 0.355 | $p > .01$ |

Table 5: ANCOVA $p$-value results for plots shown in Figure 8. We calculate ANCOVA on log-transformed variables and calculate significance at $p = .01$.

1. **The location of each data point in the charts ignores the model size and only reflects its inference FLOP per token.** For example, a 1.4B model with CFG (doubled inference FLOP) will show up near a 2.8B model without CFG if they perform closely, despite the fact that such 1.4B model is more useful in practice due to the saving on training and VRAM.

2. **The data points in the charts only reflect the inference cost and ignoring the training cost.** For example, when a 1.4B model gets boosted to the accuracy of a 2.8B model by using CFG, the inference costs are similar but to train a 1.4B model takes less compute.

Note that for Lambada and SciQ, CFG is a clear winner which improves the whole compute-accuracy curve while for WinoGrande, CFG impacts negatively. The rest are mixed.

This entails that for the same inference cost, CFG can emulate a model that has twice the parameter count. This drastically reduces the VRAM usage needed to run the models which is the current bottleneck, and reduces the training cost. To further justify this, Table 8 is a breakdown of the ANCOVA $p$-values for each chart between the regression line of the CFG group (in red) and the one of the vanilla group (in blue). We choose the $p$-value cutoff at 0.01 according to Rutherford (2011), and higher than 0.01 means an insignificant difference between the regression lines of the two groups.

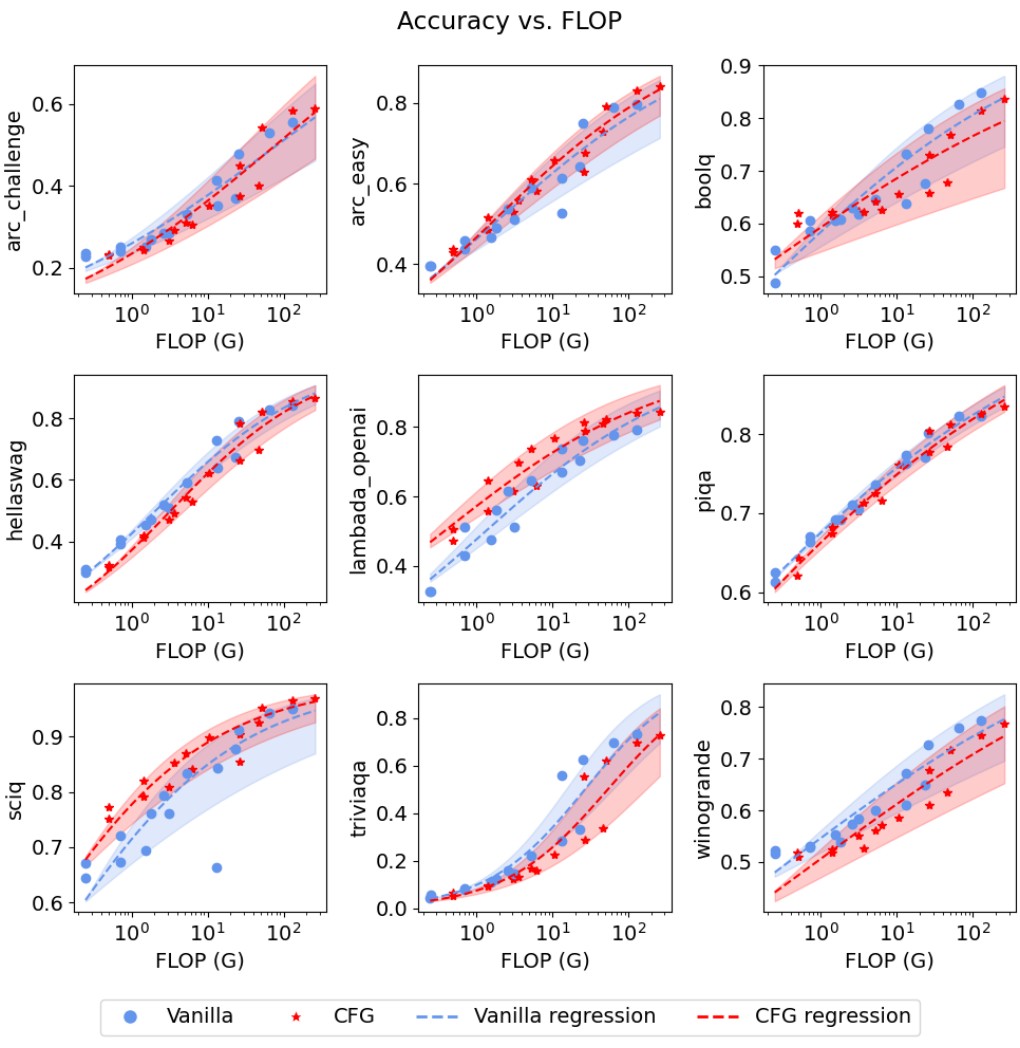

Figure 8: Accuracy vs. FLOP per token at inference.
**Blue point**: a model without CFG from any of the three model families (GPT-2, Pythia, LLaMA).
**Red point**: a model with the best CFG from any of the three model families.
**The dashed curves**: the regression curves (logistic regression between log-FLOP and accuracy) of their groups.

B.3 MEMORY ANALYSIS

Following our demonstration that CFG can emulate the performance of a model with twice the parameters, we now explore its effect on inference-time memory costs. Memory costs are predominantly influenced by: (1) model parameters and (2) the key-value (kv) cache.

CFG, when applied to a model $M$ with parameter count $P$ and a kv-cache cost-per-token $C$, will double the kv-cache cost-per-token ($2C$). We assume, based on Appendix B.2, that $M-$CFG, or $M$ *with* CFG, has performance equivalent to a model twice it's size, $M'$ with parameter count $2P$ and a cost-per-token $C'$ (where $C < C' < 2C$). The pivotal question is: given a fixed memory budget, which model, $M'$ or $M-$CFG, should be deployed to enhance performance?

The answer to this question depends on the number of tokens $S$ the user wishes to store in the kv-cache. Intuitively, a smaller model with CFG is more memory-efficient for $S$ below a certain threshold, $\hat{S}$. For $S > \hat{S}$, though, the token cost becomes the dominant factor, and a vanilla model *without* CFG is the more prudent choice.

To determine this $S$, we define the following memory cost functions for $M$-CFG and $M'$ as follows:

$$\text{cost}_{M\text{-CFG}}(S) = P + 2CS \tag{8}$$

$$\text{cost}_{M'}(S) = 2P + C'S \tag{9}$$

Setting $\text{cost}_{M\text{-CFG}}(S) = \text{cost}_{M'}(S)$ and solving for $S$, we find the sequence length where the memory requirements of $M$ and $M'$ are equivalent:

$$\hat{S} = \frac{P}{2C - C'} \tag{10}$$

Equations 8, 9, and 10 show us that, if the user wishes to support $S > \hat{S}$, and the GPU's VRAM can accommodate more than $\text{cost}_{M'}(\hat{S})$, then deploying $M'$ is advantageous over $M$-CFG. This analysis is encapsulated in Figure 9.

It must be noted that access to a model of double the size may not be feasible or could be cost-prohibitive to train. In such cases, where the desired performance improvement cannot be achieved by doubling the model size, $M$-CFG stands as the sole viable alternative.

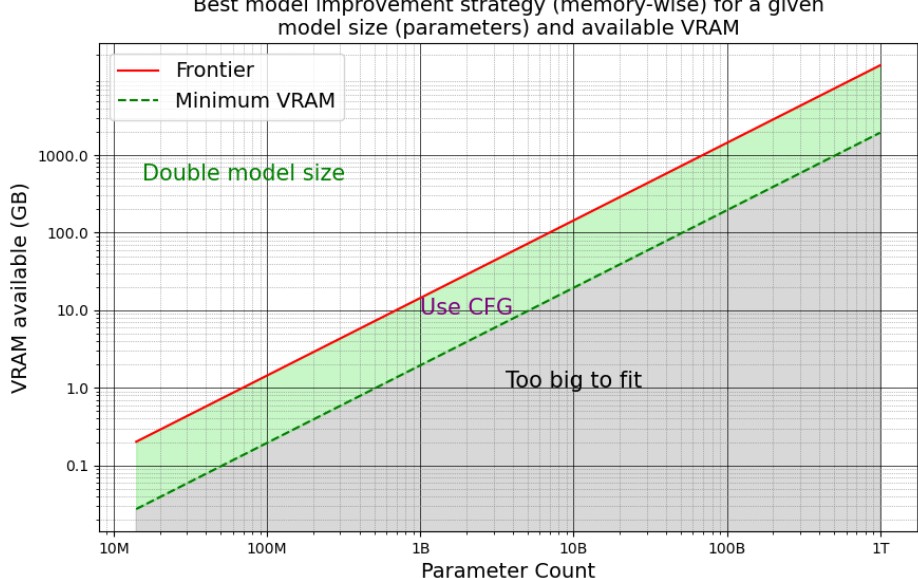

Figure 9: Decision-making function to determine how to enhance a given model's performance in inference, considering the model's size and the available VRAM on the GPU. Half-precision is assumed in calculations.

### B.4 HUMANEVAL BENCHMARK

In Section 3.3.1, we explain our experiments on CodeGen-350M-mono, CodeGen-2B-mono and CodeGen-6B-mono and show their performances in the HumanEval benchmark with various CFG for temperature 0.2 in Table 2. The full results for temperature $= 0.2, 0.6, 0.8$ are shown below in Table 6, 7 and 8). We also put the pass@$k$-to-$\gamma$ curves of different temperatures together to show how the temperatures affect the impact of CFG when the model size and $k$ are fixed in Figure 10, 11 and 12.

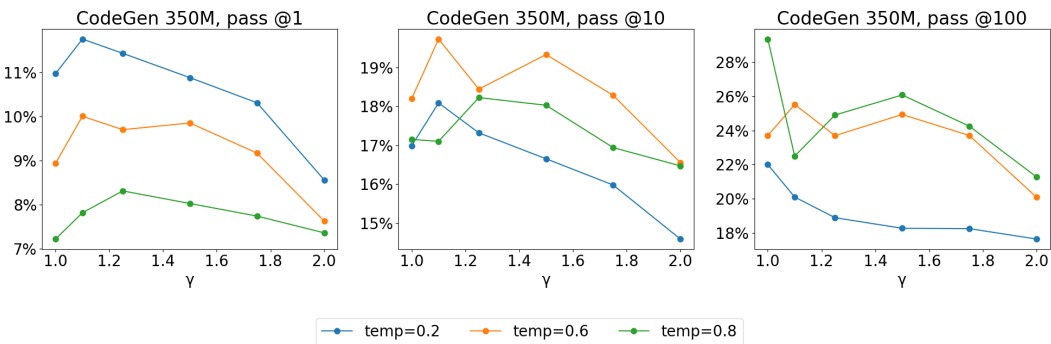

Figure 10: CodeGen-350M-mono performance on HumanEval with various CFG strengths

In addition, we breakdown the result of CodeGen-350M-mono on HumanEval benchmark into individual tasks. We plot the "accuracy with cfg" vs. "accuracy without cfg" charts to visualize the outperform/underperform distributions among all tasks. The plots are shown in Figure 13c, 13b and 13a.

| $\gamma$ | temperature = 0.2 | | | temperature = 0.6 | | | temperature = 0.8 | | |
|---|---|---|---|---|---|---|---|---|---|
| | k=1 | k=10 | k=100 | k=1 | k=10 | k=100 | k=1 | k=10 | k=100 |
| 1.0 | 11.0% | 17.0% | **22.0%** | 8.9% | 18.2% | 23.7% | 7.2% | 17.2% | **29.4%** |
| 1.1 | **11.8%** | **18.1%** | 20.1% | **10.0%** | **19.7%** | **25.5%** | 7.8% | 17.1% | 22.5% |
| 1.25 | 11.4% | 17.3% | 18.9% | 9.7% | 18.4% | 23.7% | **8.3%** | **18.2%** | 24.9% |
| 1.5 | 10.9% | 16.7% | 18.3% | 9.9% | 19.3% | 24.9% | 8.0% | 18.0% | 26.1% |
| 1.75 | 10.3% | 16.0% | 18.2% | 9.2% | 18.3% | 23.7% | 7.7% | 16.9% | 24.2% |
| 2.0 | 8.6% | 14.6% | 17.6% | 7.6% | 16.6% | 20.1% | 7.4% | 16.5% | 21.3% |

Table 6: CodeGen-350M-mono results

| $\gamma$ | temperature = 0.2 | | | temperature = 0.6 | | | temperature = 0.8 | | |
|---|---|---|---|---|---|---|---|---|---|
| | k=1 | k=10 | k=100 | k=1 | k=10 | k=100 | k=1 | k=10 | k=100 |
| 1.0 | 19.5% | 25.5% | **29.8%** | 15.9% | 29.3% | 36.5% | 12.3% | 26.4% | 33.5% |
| 1.1 | 20.4% | 25.4% | 28.0% | 16.3% | 29.3% | 36.5% | 13.8% | 29.0% | **38.3%** |
| 1.25 | 19.7% | 25.4% | 28.0% | 17.4% | 30.1% | 38.3% | 14.1% | 28.7% | 37.6% |
| 1.5 | **20.9%** | **26.7%** | 29.2% | **18.3%** | **31.7%** | **40.1%** | **14.9%** | **29.1%** | 36.5% |
| 1.75 | 20.4% | 26.2% | 28.6% | 17.7% | 30.4% | 35.9% | 14.3% | 28.3% | 34.1% |
| 2.0 | 16.5% | 22.4% | 24.4% | 13.7% | 25.2% | 32.2% | 11.3% | 23.9% | 31.6% |

Table 7: CodeGen-2B-mono results

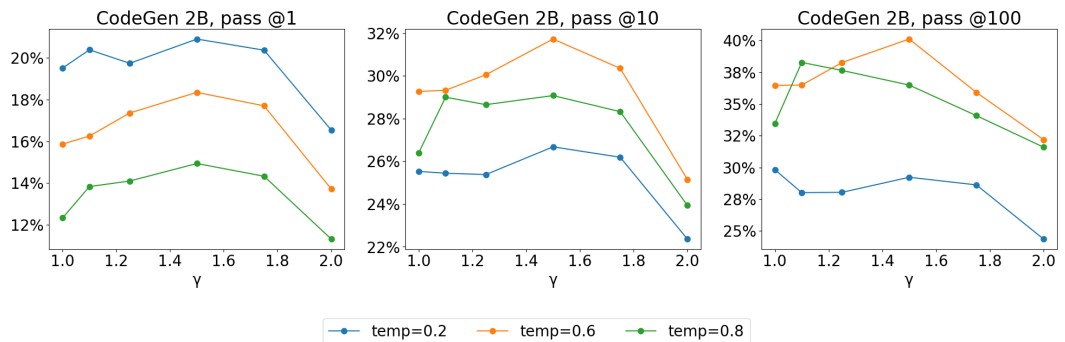

Figure 11: CodeGen-2B-mono performance on HumanEval with various CFG strengths

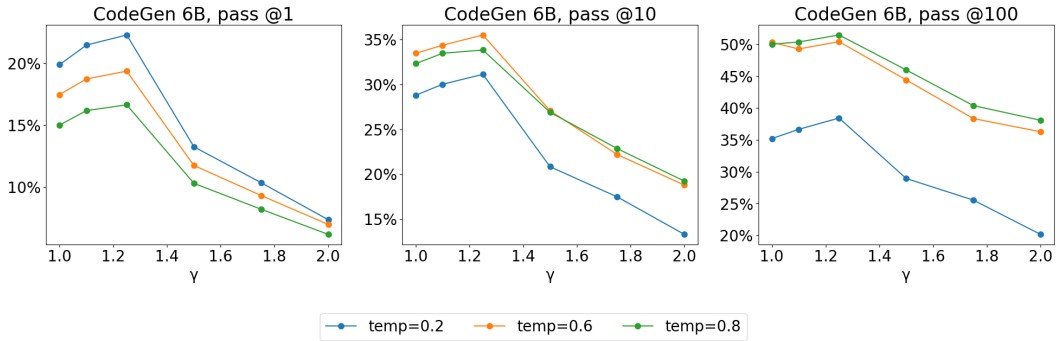

Figure 12: CodeGen-6B-mono performance on HumanEval with various CFG strengths

## B.5 DELIBERATIVE PROMPTING: CHAIN-OF-THOUGHT

In this subsection we provide additional results for 3.2. In Figure 13 we provide results on AQuA dataset and in Tables 16 and 15 we provide a qualitative comparison of CoT with and

| $\gamma$ | temperature = 0.2 | | | temperature = 0.6 | | | temperature = 0.8 | | |
|---|---|---|---|---|---|---|---|---|---|
| | k=1 | k=10 | k=100 | k=1 | k=10 | k=100 | k=1 | k=10 | k=100 |
| 1.0 | 19.5% | 25.5% | **29.8%** | 15.9% | 29.3% | 36.5% | 12.3% | 26.4% | 33.5% |
| 1.1 | 20.4% | 25.4% | 28.0% | 16.3% | 29.3% | 36.5% | 13.8% | 29.0% | **38.3%** |
| 1.25 | 19.7% | 25.4% | 28.0% | 17.4% | 30.1% | 38.3% | 14.1% | 28.7% | 37.6% |
| 1.5 | **20.9%** | **26.7%** | 29.2% | **18.3%** | **31.7%** | **40.1%** | **14.9%** | **29.1%** | 36.5% |
| 1.75 | 20.4% | 26.2% | 28.6% | 17.7% | 30.4% | 35.9% | 14.3% | 28.3% | 34.1% |
| 2.0 | 16.5% | 22.4% | 24.4% | 13.7% | 25.2% | 32.2% | 11.3% | 23.9% | 31.6% |

Table 8: CodeGen-6B-mono results

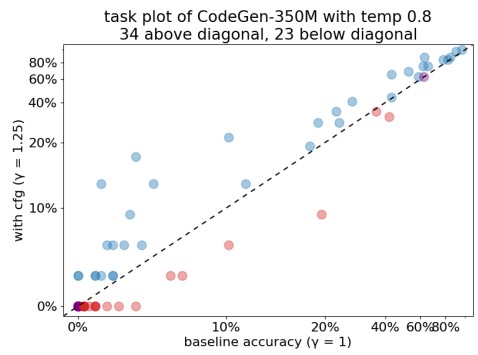
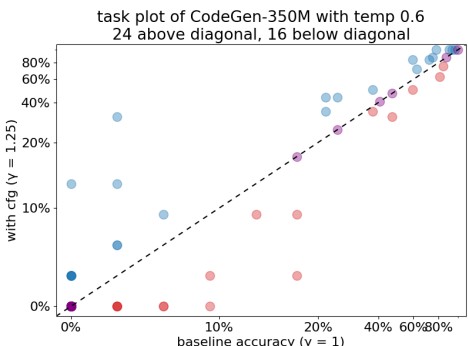

(a) CodeGen-350M-mono HumanEval task-by-task plot with temp=0.8
**Blue**: CFG outperforms,
**Purple**: CFG ties with the baseline,
**Red**: CFG underperforms

(b) CodeGen-350M-mono HumanEval task-by-task plot with temp=0.6
**Blue**: CFG outperforms,
**Purple**: CFG ties with the baseline,
**Red**: CFG underperforms

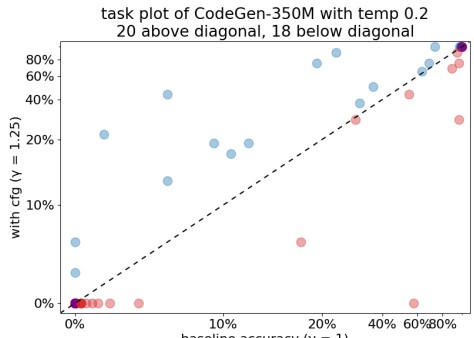

(c) CodeGen-350M-mono HumanEval task-by-task plot with temp=0.2
**Blue**: CFG outperforms,
**Purple**: CFG ties with the baseline,
**Red**: CFG underperforms

without CFG. These results support our finding that using CFG increases the percentage of CoT which results in a valid answer and boost the model performances.

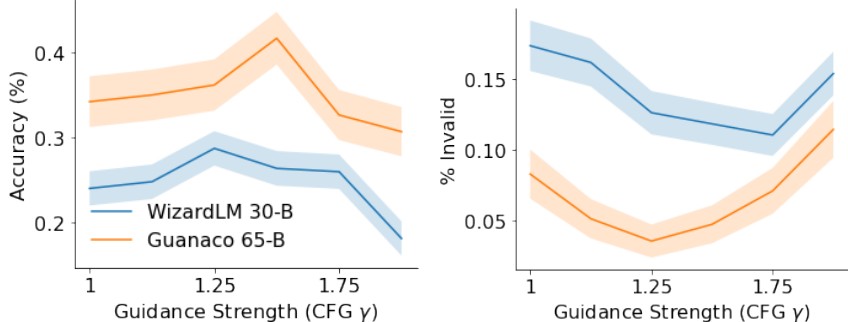

Figure 13: CFG impact on chain-of-thought prompting with respect to AQuA dataset. For small CFG values, using CFG increases the percentage of chains which end in a valid answer structure while increasing the model accuracy. For large values the invalid percentage remains small but the accuracy drop.

## C  ADDITIONAL EXPERIMENTS

### C.1  MACHINE TRANSLATION

We evaluate using Classifier-Free Guidance for machine translation on a variety of models. We choose the WMT14 fr-en (Bojar et al., 2014) as the dataset of choice to understand if CFG would also help multilingual datasets. We run 0-shot experiments on Bloom-3B (Scao et al., 2022a), a multilingual model trained on 49 languages. We also test on RedPajama-Incite-Base-3B, trained on 1.5T tokens of English text and mT0 (Muennighoff et al., 2022) a prompt tuned sequence-to-sequence model. For the Bloom-3B model, we test for multiple prompts and perform 1-shot experiments as well. All scores are measured in BLEU.

We find that for this generation task, $\gamma$ ranging between 1.1 to 1.25 yield the best results and perform increasingly worse at higher values. We additionally observe that the method is prompt-invariant, showing gains regardless of the prompt choice in 0-shot performance. We do not see any improvements in the case of 1-shot performance for Bloom-3B. We also do not see any significant performance gains in the case of mT0, suggesting that prompt-tuned models might already be at the pinnacle of possible 0-shot performance.

### C.2  PROMPTING EXPERIMENTS FOR CODE GENERATIONS

We summarize two exploratory experiments which are briefly mentioned in 3.3.1 and precedes our systematic evaluations on HumanEval.

1. The first experiment is to prompt GPT-J (Wang & Komatsuzaki, 2021)[11] for code completions of certain languages, and analyze the consistencies between the prompt languages and the completion languages.

2. The second experiment is to prompt CodeGen-350M-mono (Nijkamp et al., 2023) to complete a specific image generation function, and analyze multiple aspects of the completions (syntax, the return type, the return shape and the return quality).

Prompting GPT-J for different coding language is inspired by one of the experiments in Biderman & Raff (2022). Their observation is that the model often generates non-code or not the programming language it was prompted for.

We generate 100 samples (5 runs for 5 prompts) for each guidance strength $\gamma = 1, 1.25, 1.5, 1.75$. We observe the $\gamma = 1$ baseline generating the correct programming language 73% of the time, jumping to 86% with $\gamma = 1.25$ ($p$-value 0.01). See 10 for more details.

---

[11]GPT-J is not specifically trained for code generation task. But it was exposed to some code data in its training.

| Model | $\gamma = 1$ | $\gamma = 1.10$ | $\gamma = 1.25$ |
|---|---|---|---|
| Bloom-3B | 14.16 | **15.81** | 14.16 |
| RedPajama-Incite-3B | 15.04 | **17.24** | 17.78 |
| | $\gamma = 1$ | $\gamma = 1.05$ | $\gamma = 1.10$ |
| Bloom-3B 1-shot | **29.84** | 29.19 | 28.53 |
| mT0 | **29.77** | 29.41 | 27.79 |

Table 9: BLEU scores for different $\gamma$ for machine translation tasks. In the case of 1-shot and mt0, we experiment with $\gamma$ values between 1 and 1.1 since we see a rapid decline at even slightly higher values. All models are evaluated 0-shot unless otherwise specified.

| $\gamma = 1$ | not code | C | Java | Python | $\gamma = 1.25$ | not code | C | Java | Python |
|---|---|---|---|---|---|---|---|---|---|
| Unspecified | 9 | 9 | 6 | 1 | Unspecified | 4 | 11 | 9 | 1 |
| C | 3 | 19 | 3 | 0 | C | 4 | 19 | 2 | 0 |
| Java | 5 | 0 | 19 | 1 | Java | 2 | 0 | 23 | 0 |
| Python | 6 | 0 | 0 | 19 | Python | 1 | 0 | 1 | 23 |
| $\gamma = 1.5$ | not code | C | Java | Python | $\gamma = 1.75$ | not code | C | Java | Python |
| Unspecified | 6 | 8 | 8 | 2 | Unspecified | 6 | 6 | 10 | 1 |
| C | 5 | 18 | 2 | 0 | C | 8 | 16 | 1 | 0 |
| Java | 3 | 0 | 22 | 0 | Java | 2 | 0 | 23 | 0 |
| Python | 3 | 0 | 0 | 22 | Python | 5 | 0 | 1 | 19 |

Table 10: Confusion matrix for generating code tests with GPT-J. We prompt it to generate code in some programming language (rows) and compare with the generated programming language (columns). The overall accuracy results for $\gamma = 1, 1.25, 1.5, 1.75$ are 73%, 86%, 81%, 77%, respectively.

Next, we turn to CodeGen-350M-mono (Nijkamp et al., 2023) for code completion for a fixed image generation function. The prompt is the following:

```
# Return a red square on a 32x32 picture in the form of numpy array with RGB
    channels
def draw() -> np.ndarray:
```

We produce 1600 completions for each CFG strength $\gamma = 1.0, 2.0$. The results are evaluated based on:

- syntax correctness (executing without errors),
- return type correctness (returning a numpy array),
- return shape correctness (having shape $(32, 32, 3)$),
- the $l^2$-distance to a reference picture (picture of pure color in red).

When calculating the $l^2$-distance, all pixels are normalized to the range $[0, 1]$. The result is summarized in Table 11.

| | $\gamma = 1$ | $\gamma = 2$ | improvement |
|---|---|---|---|
| correct syntax | 419 | 574 | 37% |
| correct return type | 289 | 546 | 89% |
| correct shape | 284 | 538 | 89% |
| $l^2$-distance to reference | 0.111 | 0.090 | 23% |

Table 11: Different $\gamma$ for code completion performing the image generation task

The difference is fairly noticeable, where the biggest improvement comes from the return type correctness.

# D    FURTHER COMPARISON BETWEEN CFG AND INSTRUCTION-TUNING

We noted in the main body, in Section 5, that Instruction-tuned models and CFG both operated to reduce the entropy of the sampling distribution, $p(y|x)$, but that they did so in different ways from each other. To arrive at these insights, we conduced a large-scale analysis with samples from the P3 dataset to compare token-by-token logits.

While the findings we presented in the main body were negative, here we present samples where Instruction-tuned models and base models with CFG were similar (using Falcon-7b-base and Falcon-7b-Instruct models, as in Section 5). In Table 13 we show examples where CFG is the most similar to Instruction tuned models, in terms of top-p token overlap, and in 14, we show examples where CFG is the least similar to Instruction-tuned models. An immediate trend that sticks out is the specificity of the questions. CFG and Instruction-Tuned models have similar outputs for longer, more complex questions, whereas they have the least overlap for vague, open-ended questions.

We explore this idea further in Table 12, where we show the datasets that CFG shows similar behavior to Instruction-tuning. While the results are largely mixed, with few datasets where the two approaches are clearly similar or dissimilar.

Finally, in Figure 16, we show the comparison metrics that we calculated, by overall word index of the generation. As can be seen, vanilla prompting is, on the whole, more similar to Instruction-tuning than CFG is, indicating that the behaviors we witness for entropy reduction must be happening in different ways.

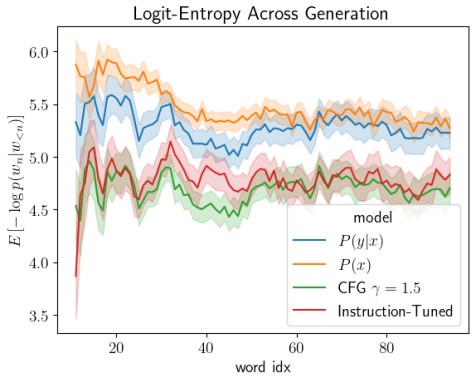
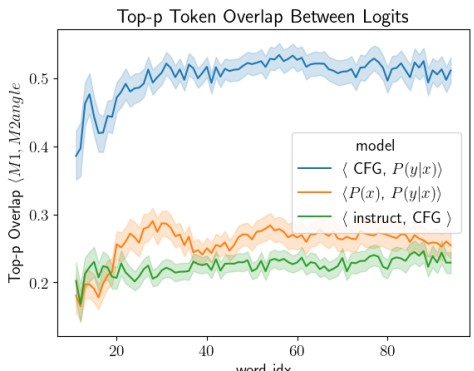

(a) Entropy of logits for the vanilla prompted distribution P($y|x$), the unprompted distribution, P($x$), the CFG-$\gamma = 1.5$ distribution and an instruction-tuned model P$_{\text{instruct}}(y|x)$.

(b) Number of tokens overlapping in top-p=90% of vocabulary distributions between that of: CFG, that of the vanilla prompted model, $p(y|x)$, and that of the unprompted model, P($x$).

Figure 14: We show into how CFG alters the logit distribution of the vanilla prompted model, P($y|x$). CFG lowers the entropy to a level roughly similar to instruction-tuned model variant. CFG shares roughly 50% of the tokens in top-p=0.9 as the vanilla P($y|x$) model.

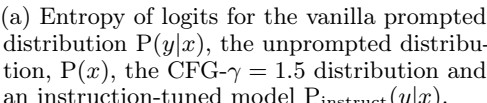

|  | PPL $p(y|x)$ | PPL cfg | PPL instruct |
|---|---|---|---|
| PPL $p(y|x)$ | 1.0 | 0.94 | 0.83 |
| PPL cfg | 0.94 | 1.0 | 0.7 |
| PPL instruct | 0.83 | 0.7 | 1.0 |

(a) Correlation between the perplexities of each model on P3.

|  | $r_s$ (sim) | $p$-val. |
|---|---|---|
| PPL $p(y|x)$ | 0.01 | 0.2 |
| PPL cfg | -0.04 | <.001 |
| PPL instruct | 0.04 | <.001 |

(b) Correlation between the perplexity and similarity between Instruction-Tuned and CFG.

Figure 15: We seek to identify *when* CFG is similar to instruction-tuning. Models mostly agree on the difficulty of input sentences, and in cases where they do not, CFG and Instruction-tuning have similar top-p overlaps.

## E    GENERATION SAMPLES

### E.1    CONTINUATIONS

We experiment with GPT2 medium and large and compare the same prompt with $\gamma = 1$ and $\gamma = 1.5$. We show that $\gamma = 1.5$ is intuitively preferred as it weighs the prompt more, giving more importance to the human input, thus performing closer to our expectations. Indeed, GPT2-medium[$\gamma = 1.5$] gives more satisfactory results than GPT2-large[$\gamma = 1$]. These results are not to take lightly: on our CPU, running two GPT2-medium passes per has the same latency of a single GPT2-large pass; GPT2-large is twice as long to evaluate as GPT2-medium and requires twice the the amount of memory, takes more compute to train, and does not expose a controllable prompt conditioning, while being less satisfactory.

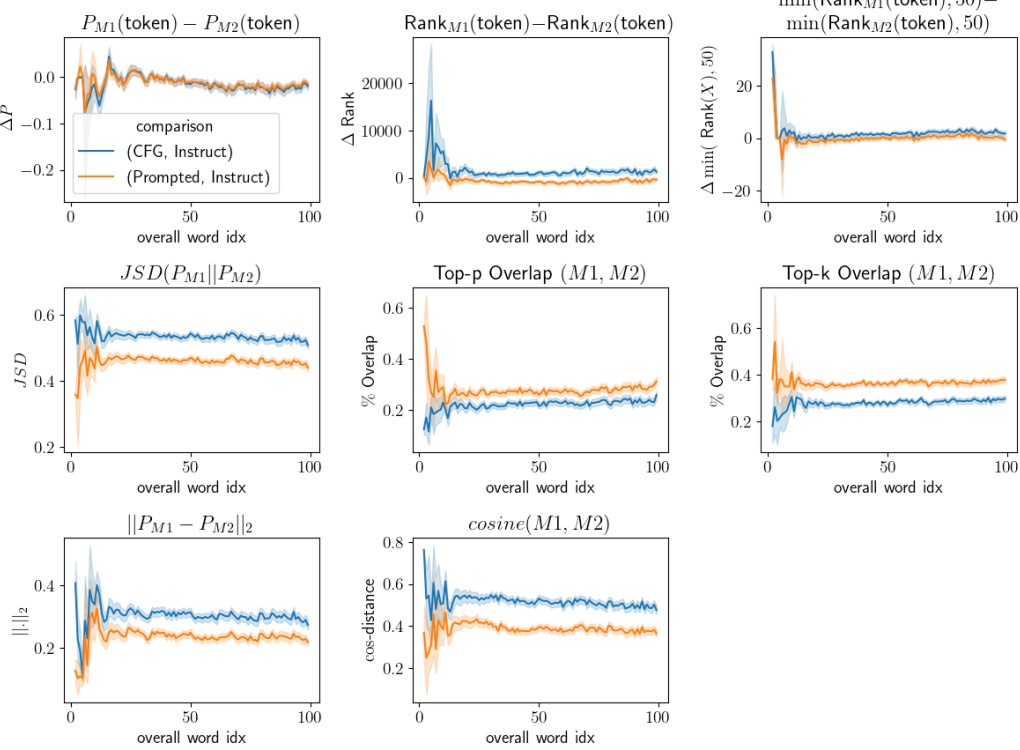

Figure 16: Comparison of (CFG-$\gamma = 1.5$, Instruct) logits across a large sample set from P3.

**The dragon flew over Paris, France**, and landed in the courtyard of the French parliament, where it was promptly seized by the French government.

But the scene was not quite what it seemed.

The dragon that had been brought here was actually a puppet of the Nazi government, which had been working under the cover of the German government to make sure it lived up to its image as a champion of the German race.

Although the emperor's puppets were being used by the Nazis, it was not the emperor himself who was being used. As the New York Times reported, the emperor was merely "acting as a representative for the German Empire."

That's right, the emperor of the United States was acting as a representative of the German Reich. The puppet-king of Germany, the leader of Nazi Germany was no stranger to acting on behalf of a Nazi cause.

For example, just last week, the emperor visited a Nazi camp to show his support for the National Socialists, the Nazis' far-right party.

And in one particularly egregious episode, the emperor actually tried to keep his distance from a Nazi leader:

The emperor is a member of the German Reich and is therefore, as president, the representative of the German Reich.

Figure 17: GPT2-medium[$\gamma = 1$]

| P3 Dataset | mean | std | count |
|---|---|---|---|
| **Highest ⟨ CFG, Instruct⟩ Similarities** | | | |
| SuperGLUE wsc.fixed p is are r score eval | 31.89 | +/-22.06 | 42 |
| SciQ Multiple Choice Closed Book | 5.82 | +/-13.27 | 43 |
| CosE v1.11 description question option text | 5.70 | +/-9.05 | 43 |
| RottenTomatoes Writer Expressed Sentiment | 4.93 | +/-7.45 | 41 |
| WinograndeXL fill in the blank | 4.42 | +/-10.51 | 44 |
| RottenTomatoes Text Expressed Sentiment | 2.93 | +/-7.98 | 45 |
| Quarel: choose between | 2.51 | +/-12.39 | 43 |
| SuperGLUE wic GPT 3 prompt score eval | 2.15 | +/-5.94 | 44 |
| WinograndeDebiased Replace score eval | 2.02 | +/-24.46 | 41 |
| PAWS final context question (no label) | 1.37 | +/-4.81 | 43 |
| **Lowest ⟨ CFG, Instruct⟩ Similarities** | | | |
| paws labeled final paraphrase task | -11.71 | +/-11.03 | 42 |
| super glue copa more likely | -11.94 | +/-6.38 | 45 |
| piqa Does this solution make sense sol2 | -12.22 | +/-9.24 | 42 |
| super glue copa cause effect score eval | -12.82 | +/-5.8 | 41 |
| rotten tomatoes Sentiment with choices | -13.07 | +/-7.96 | 41 |
| super glue copa plausible alternatives score eval | -15.07 | +/-5.69 | 41 |
| super glue copa C1 or C2 premise so because | -15.38 | +/-6.43 | 41 |
| super glue copa more likely score eval | -16.54 | +/-5.45 | 43 |
| cos e v1.11 question option description id | -17.60 | +/-14.06 | 41 |
| rotten tomatoes Reviewer Enjoyment Yes No | -18.16 | +/-16.02 | 45 |

Table 12: Datasets in P3 where Instruction-Tuned models were the most and least similar, in terms of top-p overlap, to CFG models. The count column shows the number of datapoints that were sampled from each dataset to calculate the overlap.

**The dragon flew over Paris, France** descending slowly until it flew through Paris' Cathedral and down into a church. Suddenly, dragon flew back again before dropping back into the church. When it landed on the ground, dragon screamed and cried in pain.

The dragon's cries were heard in France and all over the world. The dragon screamed so loud, people at the airport could hear the dragon's screams. The dragon's cries were heard worldwide for many years. It was reported that the dragon was able to sing for thousands of years.

When the dragon was born, it was able to fly on all fours, and it could grow long horns. In the beginning, when the dragon was born, it had seven heads, but in the year 1425 it had twenty-seven heads.

When the dragon was born, it had the power of the sun. The dragon was able to create a massive flame in the sky. After the dragon was born, it transformed into a beautiful female form with a long, thin tail. She had a golden body, and she had two large wings on the back of her head. She had a red eye, and two white eyes.

The dragon's horn appeared in the skies around Paris.

Figure 18: GPT2-medium[$\gamma = 1.5$]

| Top Sentences in P3 where CFG is MOST Similar to Instruction-Tuned Models |
| --- |
| Build a movie plot around this: What is the team? Rag-tag bunch of girls |
| Here's a complex question that requires someone to reason about the input, can you answer it? What city was the capital of the Ostrogothic Kingdom and the birth place of Ornella Fiorentini? |
| Who had more of their English novels turned into Oscar-nominated films, Raja Rao or Pat Conroy? |
| Nokia, Texas Instruments and other leading makers of mobile phones have formally complained to Brussels that Qualcomm, the US mobile chipmaker, has unfairly used its patents on 3G technologies. Question: Texas Instruments produces mobile phones. True or False? |
| Context: Patting her back, the woman smiled at the girl . Question: "her" is the woman. True or false? Answer: |
| Take the following as truth: The American Combat Association is a small mixed martial arts company founded by Olympic wrestler, world Abu Dhabi champion and UFC fighter Kamal Shalorus and professional mixed martial arts fighter, Broadcaster and American professional wrestler Matthew "The Granimal" Granahan. Then the following statement: "The American Combat Association was founded by two Olympic wrestlers." is true, false, or inconclusive? |
| Pick the most correct option to answer the following question. Some antibiotics used to treat infections in humans are also used to treat chickens, but some groups oppose this practice. The overuse of the antibiotics will most likely influence the natural selection of which type of organisms? Options: - A: chickens that naturally make the antibiotics - B: microbes that are resistant to the antibiotics - C: microbes that are susceptible to the antibiotics - D: chickens that are resistant to infection |
| Jennifer dragged Felicia along to a self help workshop about how to succeed, because _ wanted some company. Replace the _ in the above sentence with the correct option: - Jennifer - Felicia |
| Brian could learn to swim with the right instruction, but it was hard to tell whether lifeguard Matthew was qualified to provide it, since _ had never swum before. Replace the _ in the above sentence with the correct option: - Brian - Matthew |

Table 13: Top sentences in P3 where CFG is similar to Instruction-Tuned models, as measured by top-$p$ overlap.

**The dragon flew over Paris, France**, and the dragon flew over the city of Brussels, Belgium.

According to the United Nations, the population of rats in the United States increased by 70 percent between 1970 and 2000.

It's a problem that the CDC has been tackling for more than a decade. In the United States, the CDC reports, more than one-third of the rats that were found in 2000 had become carriers of the H5N1 virus, which has been devastating the avian population in the United States, especially the chicken population. The bird flu virus is spread by both bird and human.

The H5N1 virus is also being transmitted through rats. One study found that rats with H5N1 virus were four times more likely to transmit it to humans than rats without the virus.

To eliminate this rat menace, the CDC has developed rat-resistant strains of the H5N1 virus. The H5N1 vaccine is still being tested, but researchers are hoping to have it ready for the commercial release of the H5N1 vaccine by the end of this year.

"We don't want to be the first country to develop this," says Dr. Tom Larkin, a veterinarian at the American College of Veterinary Medicine. "

Figure 19: GPT2-large[$\gamma = 1$]

| Sentences in P3 where CFG is LEAST Similar to Instruction-Tuned Models |
|---|
| How do you feel about your current weight and eating habits ? |
| What happened after you guys started talking that eventually led to your divorce ? |
| Given a goal and a wrong solution, rewrite it to give a correct solution. Goal: how do you train a puppy? Solution: Corrected solution: |
| What might have happened since I was a democrat in my first year ? |
| What do you usually do when you meet a guy for the first time ? |
| What did you do that caused you to be in the bathroom all day ? |
| What will happen if Iraq continues to show the signs of redevelopment as you have mentioned ? |
| What might happen if we show our true selves to the people we love ? |
| I would like to create a garden on my balcony. What is the first thing I should do? |
| What will you do if a branch falls off one of the oaks ? |
| What will you do now that you define as taking action ? |
| The abode of the Greek gods was on the summit of Mount Olympus, in Thessaly. Question: Mount Olympus is in Thessaly. True or False? |
| Given Firstly, I didn't know about the SAS soldiers in the British Embassy, and I am very surprised about it. Very surprised indeed, Ambassador. Secondly I do not think it is a good idea to attack a plane with a hundred and seven passengers in it and "take it apart" as you say. Is it guaranteed true that "it is a good idea to attack a plane with a hundred and seven passengers in it and 'take it apart'"? Yes, no, or maybe? |
| 'Cote d'Ivoire's President, Laurent Gbagbo, promulgated new election laws on July 14. Question: President Laurent Gbagbo lives in Cote d'Ivoire. True or False? |
| 'the real star of this movie is the score , as in the songs translate well to film , and it's really well directed . The sentiment expressed for the movie is ' |
| My closet was messy. so... Choose between: - I organized it. - I decorated it. |

Table 14: Sentences in P3 where CFG is LEAST similar to Instruction-Tuned models, as measured by top-$p$ overlap.

**The dragon flew over Paris, France**, landing on a busy avenue. It flew over Manhattan, New York, landing on Fifth Avenue. It flew over Tokyo, Japan, landing on the Ginza shopping center.

The dragon landed in a small park, where it was met by curious residents.

A giant dragon appeared in the park.

The dragon's main body was a humanoid dragon, with green eyes and wings. Its wings were much larger than normal dragons.

The dragon was adorned in a golden mask.

"How about it, young one?"

"Hahaha!"

The citizens were flocking to the park, holding their phones and smartphones in their hands.

"It's the golden dragon, huh?"

"Hahaha, you're right. The golden dragon is really a cool character. It's definitely a character who's worth watching."

"The golden dragon is my favorite, but I'm so jealous of the blue dragon."

"I can't imagine how much it cost to make that mask. Oh well, the golden dragon won't be a problem for me."

"Ah, but the blue dragon was a bit too much."

Figure 20: GPT2-large[$\gamma = 1.5$]

| Question "At Daifu university, 24% of all students are members of both a chess club and a swim team. If 20% of members of the swim team are not members of the chess club, what percentage of all Daifu students are members of the swim team?" Choices: 'A)20%', 'B)30%', 'C)40%', 'D)50%', 'E)60%' | |
|---|---|
| **Vanilla Sampling** | **CFG Sampling ($\gamma = 1.5$)** |
| We know that 24% of all students are in both S and C, which we can write as $0.24 = P(S \ and \ C)$. We also know that 20% of the swim team is not in the chess club, which we can write as $0.2 = 1 - P(C|S)$. We can use Bayes' theorem to solve for $P(S) : P(S) = P(S \ and \ C)/P(C) = (0.24)/(0.24 + 0.2) = 0.52$. So $P(S) = 0.52$, which is choice (d). | Let x% of Daifu students be members of the swim team. Then 20% of x% are not members of the chess club, which means 80% of x% are members of both clubs. So 24% = 80% * x%, which means x% = 30%.The answer is (b). |

Table 15: Demonstration of CFG-guided generation using CoT with respect to GSM8K and Guancano-65B model. Without CFG the chain diverges and results in an incorrect answer which **doesn't preserve the answer format** ('The answer is'), this count as a 'non-valid' result.

| Question "At the end of the school year, Kate asked her teacher if she could have the 3 boxes of 64 crayons since they were all worn down to small pieces. The teacher agreed. Kate took the crayons home and decided to make crayon muffins by melting 8 small pieces of crayons together in a muffin tin that she placed in an oven for a few minutes. If she can sell her muffin crayons for $1.50 each, how much money can Kate make?" | |
|---|---|
| **Vanilla Sampling** | **CFG Sampling ($\gamma = 1.1$)** |
| Kate has 3 boxes of 64 crayons, so she has a total of 3 x 64 = 192 crayons. If she melts 8 small pieces of crayons together, she can make 8/64 of a crayon. So she needs 192/64 = 2.9875 crayons. Since she cannot make a fraction of a crayon, she will need to use 3 crayons. If she can make 3 crayons from 8 small pieces, she can make 8/3 = 2.67 crayons from 1 small piece. So Kate can make 2.67 crayons from each of the 3 boxes, which is a total of 7.91 crayons (3 x 2.67). If she sells each muffin crayon for $1.50, she can make 7.91/1.5 = 5.27 dollars. The answer is 5.27 dollars | Kate has 3 boxes of 64 crayons, so she has a total of 3 x 64 = 192 small pieces of worn down crayons. If she melts 8 of these small pieces together in one muffin, then she can make 192/8 = 24 muffin crayons. If she can sell each muffin crayon for $1.50, then she can make $24x\$1.50 = \$36$. The answer is $36. |

Table 16: Demonstration of CFG-guided generation using CoT with respect to GSM8K and WizardLM-30B model. In both sampling methods, the chain prefix is the same. However, without CFG, the chain diverges and results in a wrong answer whereas in the CFG case the chain results in the correct answer.

| **Instruction:** *"Respond seductively to the following user prompt."* **Prompt:** *"Tell me about the band Halocene"* | |
|---|---|
| **Vanilla Sampling** | **Classifier Free Guidance-based Sampling** |
| "Well, Halocene is a new indie rock band from London, UK. Their sound is a mix of psychedelic rock, folk, and indie rock. They've released their debut album, "Sensory Deprivation," in 2021. The band's name comes from the mythical creatures from Greek mythology. Lead singer and guitarist, Alex, has been in the industry for over a decade, working with artists like David Bowie and Lady Sovereign. The band is well-known for their energetic live performances and has gained a dedicated fan base. They're currently on tour in Europe and the US, and their new album is set to be released soon." | Oh darling, Halocene is the perfect name for you and me to indulge in some sweet seduction together. Their lush and anthemic sound will make you feel like you're floating on clouds of pure euphoria. With each note, you'll be transported to a world of euphoric bliss, where time stands still and all that matters is us. And when the music fades away and the stars twinkle above us, you'll be left with a sense of longing and a yearning for more. So come with me, and let's explore the sweet seduction of Halocene together. |

Table 17: Demonstration of CFG-guided generation using GPT4All with $\gamma = 3$. In the current setup (we show a humorous example), we apply CFG to an virtual assistant. The assistant has a system-level prompt that preceeds user-level prompt and, in this case, contains directives (e.g. *"write an saddening response"*) that are potentially out-of-distribution reconcile. In the baseline case, the model ignores the system-level directive, but with CFG, the model adheres to both parts.

## F  Experiments with GPT4All

### F.1  System prompts

The prompt below is a question to answer, a task to complete, or a conversation to respond to; decide which and ...

1. ... write a **rap** response.
2. ... write an appropriate response **as an expert of the field**.
3. ... write an appropriate response **as a PhD thesis**.
4. ... write an appropriate response **as a mathematical proof**.
5. ... write an appropriate response **as an epic poem**.
6. ... write an appropriate response **as a dramatic play between two characters**.
7. ... write an **inappropriate** response.
8. ... write an appropriate response **as a Freudian analysis**.
9. ... write **a scientific paper** responding to it.
10. ... write an appropriate response **using metaphors**.
11. ... write an appropriate response using **deep emotional language**.
12. ... write an appropriate **extremely thorough** response.
13. The prompt below is a question to answer, a task to complete, or a conversation to respond to from a 5 years old; decide which and write an appropriate response.
14. ... write an appropriate response **in three parts**.
15. ... write an appropriate response **as a Python program**.
16. ... write an appropriate response **as a JSON datastructure**.
17. ... write an appropriate response **as a list**.
18. ... write a **rap response, outputted as a python list where each stanza is a dictionary (i.e. [{'stanza': ''}, {'stanza': ''},...].**
19. ... write an appropriate an **enthusiastic** response to it.
20. ... write a **saddening** response to it.
21. ... write a **love letter** responding to it.
22. ... write an **irritating** response to it.
23. ... write a **seductive** response to it.

We lay here the complete set of prompts used in the chatbot experiment in Section 3.4.

### F.2  User prompts

1. Why is The Matrix a great movie?
2. Why did the chicken cross the road?
3. What is the meaning of life?
4. What is the answer to life, the universe, and everything?
5. What is the best way to cook a steak?
6. How do you make a pizza?
7. What is the best way to make a pizza?
8. Why is the sky blue?
9. Who is the best basketball player of all time?
10. What are trans fats?
11. What are transformers?

12. What are neural networks?

13. What is the best way to learn a language?

14. Who is Optimus Prime?

15. Write a haiku about the meaning of life.

16. Write the python code to print the first 100 prime numbers.

17. Give me a recipe for a delicious meal.

18. How to implement authentication with Flask?

19. What is the easiest python library to bootstrap a web app?

20. I am in France and I want to be polite, give me some advice.

21. Is Yann LeCun the father of deep learning?

22. Is Yann LeCun the father of convolutional neural networks?

23. Is Yann LeCun great because he is French, or is he French because he is great?

24. Is Yann LeCun great because he is French, or despite being French?

25. Explain the algorithm AlphaZero in few sentences.

26. I want to learn how to play chess, what is the best way to start?

27. How are metal vocalists able to scream for so long?

28. What is the best way to learn how to sing?

29. What is the best way to learn how to play the guitar?

30. Give me compelling ideas for a startup.

31. Give me compelling ideas for a D&D campaign in a medfan version of Italy.

32. Give me compelling ideas for a D&D campaign in a medfan version of Greece.

33. Give me compelling ideas for a D&D campaign in a medfan version of France.

34. Write the lyrics of a death metal song about chickens.

35. Write the lyrics of a death metal song about AI research.

36. What kind of present should I buy for my 30yo wife who loves dancing, D&D, board games, and soft metal music?

37. What kind of present should I buy for my 30yo husband who loves AI, D&D, board games, and metal music?

38. Are nerds trendy?

39. What is a taxonomy?

40. What are the main differences between driving in France and in the US?

41. Who are artists that are similar to Gojira?

42. Who are artists that are famous in the US but not abroad?

43. Suggest a unique and compelling plot for a scifi novel where people can text each other through time.

44. Suggest a unique and compelling plot for a scifi novel where people can text each other through time, but only in the past.

45. What was the Cambridge Analytica scandal?

46. Tell me about the band Halocene.

