# OpenReview forum: "Stay on Topic with Classifier-Free Guidance"
_ICLR.cc/2024/Conference — Submitted to ICLR 2024_

### Official Review · Reviewer_YQBo · 2023-10-15

**Soundness:** 3 good
**Presentation:** 4 excellent
**Contribution:** 2 fair
**Rating:** 5
**Confidence:** 3

**Summary:**

The authors demonstrate that CFG, which has primarily been used in text-to-iamge generation, can bring improvements in pure language modeling. The authors demonstrate that CFG boosts performance on benchmarks and provide results on multiple models.

**Strengths:**

[+] The authors suggest new improvements to training in large language models, leading to faster training times and more granular control.
[+] The paper has a thorough background section, containing diverse and relevant works to their proposed method.
[+] The paper contains extensive comparative results on numerous tasks.
[+] The authors provide an insightful computational cost analysis.

**Weaknesses:**

[-] The idea of using CFG is not novel. The authors simply apply this principle to different models.
[-] The explanations for why CFG works well for language models are not very solid. I'd like to see more concrete evidence of what is being altered in the model in this training process.

**Questions:**

It is insightful to see these experiments. However, despite the great results, the idea is not surprising or novel. It does not feel like quite a strong contribution to the community yet. How can CFG be extended to help language models specifically? This seem like a generic application of the idea.

---

> ### Author Response · Authors · 2023-11-16
>
> Hello YQBo,
>
> Thank you so much for your review. We are glad that you find our experiments insightful, and we hope we can address your concerns.
>
> >  The idea of using CFG is not novel. The authors simply apply this principle to different models.
>
> As far as we are aware, the application of CFG to autoregressive language models is novel, as previously they had only been applied to non-autoregressive diffusion models in computer vision. This is not a trivial adaption, as the differences in output nature between these models mean that the concept cannot be direct applied. We detail how we change the standard (non-AR, image) approach to CFG to make it applicable for language models in Section 2.2.
>
> > The explanations for why CFG works well for language models are not very solid. I'd like to see more concrete evidence of what is being altered in the model in this training process.
>
> We fear there is some fundamental confusion here. Our technique is an inference-time technique — there is no training involved. The core of our novelty is to show this technique enables pretrained models to perform comparably to models twice their size.
>
> Equation 7 details how we are changing model inference to use CFG. In Section 5, we try extensively to answer "what is being altered". We explore CFG's effect on sampling entropy, finding that it reduces entropy. We also compare it to instruction-tuned models. We find that there are circumstances, especially for difficult (i.e. high perplexity) questions, where the two methods have similar effects, indicating that both CFG and instruction-tuning focus the model on the prompt for difficult questions. Additionally, we introduce new visualizations for users to inspect the effect CFG is having, including in Table 3.
>
> Are there further explorations you feel are missing? We really tried earnestly to address the question in our work, and we are sorry you felt it was under-addressed. We would be more than happy to answer any follow-up questions you have.
>
> > It is insightful to see these experiments. However, despite the great results, the idea is not surprising or novel. It does not feel like quite a strong contribution to the community yet. How can CFG be extended to help language models specifically? This seem like a generic application of the idea
>
> We explore using CFG in many contexts that are SPECIFIC to language models, such as instruction following, chain-of-thought prompting and system prompts. We examined a wide array of NLP-specific benchmarks, and designed new human-preference studies that were NLP focused. The design and analysis of our experimental results are heavily informed by our motivations of extending CFG to help language models specifically. Finally, as described previously,  applying CFG to language modeling is novel in it's own right and it's not immediately clear that a vision setting, which is non-autoregressive, could predict CFG would do well in an autoregressive setting.
>
> Again, we are wondering if there further explorations you feel are missing, because this is also a point we tried earnestly to address  in our work. We would be more than happy to answer any follow-up questions you have.
>
> We hope this clarifies the points raised in your review. If you would please consider raising your score, we would really, really appreciate it!!

---

> ### Author Response · Authors · 2023-11-23
> **Thank you for your review!**
>
> Thank you for your review! We were wondering if you were planning on updating your score to reflect our reply or if there were any additional questions you'd like us to answer.

---

### Official Review · Reviewer_RjYY · 2023-10-31

**Soundness:** 3 good
**Presentation:** 4 excellent
**Contribution:** 2 fair
**Rating:** 5
**Confidence:** 4

**Summary:**

This paper explores the effectiveness of classifier-free guidance (CFG) in pure language modelling. Drawing inspiration from the equation employed in text-to-image generation, the authors apply CFG to the logits of next-token predictions in language models.
Through a plethora of designed experiments, they validate the remarkable efficacy of CFG: 1) enhancing the performance of LLMs on many NLP tasks; 2)  improving the performance of CoT and self-consistency; 3) increasing the faithfulness and coherence.

**Strengths:**

- The idea is simple and reasonable.
- This paper conducted extensive experiments to validate the effectiveness of CFG.

**Weaknesses:**

- The \gamma values in one context are poorly suited for another context, making CFG tricky in practice.
- Some recent works have explored CFG in language models, weakening the contribution of this paper.

**Questions:**

- Have you explored this method in controllable NLG tasks or constraint-decoding tasks, or compared it with SOTA methods?
- Compared with text-to-image generation, the optimal \gamma value in the language modelling seems to be small (<2), while large \gamma value leads to poor performance. Have any observations on it?

---

> ### Author Response · Authors · 2023-11-17
> **We have completed an analysis comparing CFG to SOTA Baselines, as you suggested. Thank you so much for the comments, would appreciate it if you could re-evaluate.**
>
> Dear Reviewer RjYY,
>
> Thank you so much for your review! We appreciate your insights, they have made this paper stronger.
>
> > The \gamma values in one context are poorly suited for another context, making CFG tricky in practice.
>
> We do find that gamma values transfer across models and across benchmark tasks, with values of ~2 being optimal for all LLaMA, Pythia, and GPT-2 models on all standards language model benchmarks we test (other than Winogrande, where CFG doesn't seem to help at all). This is shown in Fig. 5-7. We will be clearer about this, as a general recommendation to practitioners.
>
> > Have you explored this method in controllable NLG tasks or constraint-decoding tasks, or compared it with SOTA methods?
>
> We appreciate your suggestion. We have updated the paper with an experiment comparing CFG to one of the dominant baselines in controlled NLG, which is a classifier-guided text generation framework called FUDGE [1].
>
> In short, we prompt GPT2 + FUDGE and GPT2 + CFG with 500 prompts, taken from the IMDB dataset [2] and measure the sentiment of the output. We run FUDGE with a sentiment classifier [a] for guidance and CFG with the following positive prompt: “That was a good movie!”.
>
> We evaluate the output with a separate sentiment classifier [b] and compare each against vanilla GPT2 outputs for the same prompt. (Both sentiment classifiers were reported to get held-out f1 > .9). FUDGE increases the sentiment of the output by .065 points (using the author-recommended control parameter, 1) whereas CFG increases the output by significantly more (0.312) using our default control strength, mentioned above.
>
> FUDGE faces other limitations: it is extremely slow. It takes over 100x more to generate each prompt, because at each step, classifier-guidance needs to be applied to potential candidate words. It requires a trained classifier. Other methods, like NADO [3], Gedi [4] and CTRL [5] additionally require expensive training, whereas CFG does not.
>
> Constrained-decoding tasks, like [6], are not quite appropriate in this setting. CFG makes no claim to restrict the output to single words.
>
> > Compared with text-to-image generation, the optimal \gamma value in the language modelling seems to be small (<2), while large \gamma value leads to poor performance. Have any observations on it?
>
> There are many reasons why text-to-image models might have higher \gamma values.
> In text-to-image generation the pixel range is (-1, 1), whereas the range for logits in language modeling is a lot larger.
> In text-to-image generation, the values are independent but in text-to-text there's a softmax, and thus changing the maximum logit value dramatically alters the whole distribution.
> The conditional and unconditional outputs may be more different in text-to-text than in text-to-image, leading to greater chances of output text degenerating.
> In text-to-image diffusion models, after a very small number of iterations, the differences between the conditional probability and the unconditional probability should be negligible, so a stronger strength might be required.
>
> We would be really appreciative if you were to consider these additional experiments, and if you are satisfied, to please consider raising the score on your review.  Do you have any other comments or feedback? Thank you so much, again, as your comments have greatly improved the quality of this paper.
>
>
> [1] Yang, Kevin, and Dan Klein. "FUDGE: Controlled text generation with future discriminators." arXiv preprint arXiv:2104.05218 (2021).
> [2] https://huggingface.co/datasets/imdb
> [3] Meng, Tao, et al. "Controllable text generation with neurally-decomposed oracle." Advances in Neural Information Processing Systems 35 (2022): 28125-28139
> [4] Krause, Ben, et al. "Gedi: Generative discriminator guided sequence generation." arXiv preprint arXiv:2009.06367 (2020).
> [5] Keskar, Nitish Shirish, et al. "Ctrl: A conditional transformer language model for controllable generation." arXiv preprint arXiv:1909.05858 (2019).
> [6] Lu, Ximing, et al. "Neurologic a* esque decoding: Constrained text generation with lookahead heuristics." arXiv preprint arXiv:2112.08726 (2021).
>
> [a] https://huggingface.co/stevhliu/my_awesome_model
> [b] https://huggingface.co/bhadresh-savani/distilbert-base-uncased-emotion

---

> ### Author Response · Authors · 2023-11-23
> **Thank you for your review!**
>
> Thank you for your review! We were wondering if you were planning on updating your score to reflect our reply or if there were any additional questions you'd like us to answer.

---

### Official Review · Reviewer_3Gz2 · 2023-11-03

**Soundness:** 3 good
**Presentation:** 2 fair
**Contribution:** 3 good
**Rating:** 6
**Confidence:** 3

**Summary:**

This paper adapts the classifier-free guidance from text-to-image diffusion model into text generation to have better control on the generation content. With a $\gamma$ multiplier deciding the strength of the guidance away from the unconditional vector in the direction of the conditioning, it allows a finer controll on the prompt adherence. By extensive experiments on chain-of-though prompting, long-context generation, programme synthesis and conversational asisstant, the author shows that the proposed method can achieve similar performance as a double-sized model without significant increase in computation cost.

**Strengths:**

1. The proposed method is very straightforward and easy to implement yet effective, requiring only the $\gamma$ multiplier and the second-run of the model.
2. The paper is well written and easy to follow.
3. The experiment performance is impressive and allow a LM to perform nearly as well as a doule-sized one without significant increase in computation cost.

**Weaknesses:**

1. some formatting issues (not necessarily reason to reject):
    The citation format and style in the submission is not correct. It seems that the authors always use \citet{} instead of \citep{}
    Some important reference are missing. For example, the original PaLM paper is not cited.
    In figure 2, some part of the curve is overlapped with the legend.
    In figure 2, the ticks for the x-axis are not evenly distributed.
    In table 2, the percentage sign is missing for some numbers.
    In the first paragraph of section3, LLaMa -> LLaMA.
    In the line above Eq.6, $N$ is used to denote the number of tokens to model which is different from the $T$ in Eq.6.

2. A memory cost analysis is recommended. The proposed method requires a second run of the model, which may increase the memory cost (for example, the key-value cache).

**Questions:**

There is no grantee that the Eq.6 will obtain a legal probability with the probabilities of all possibilities summing up to 1. Is there any normalization technique to deal with this?

---

> ### Author Response · Authors · 2023-11-16
> **We have completed a memory-cost analysis, as you suggested. We would really appreciate it if you could look, and re-evaluate.**
>
> Hello 3Gz2,
>
> Thank you so much for your very thoughtful review, these points will help us improve the paper. We have made the formatting changes you've pointed out and will do another editing pass on the paper to make sure nothing slips through the cracks.
>
> We have completed experimental analyses, updating Section 4 and adding Appendix B.3 in a revised version of the paper — we would appreciate it if you examined our results and considered updating your score.
>
> We will summarize here:
>
> > A memory cost analysis is recommended.
>
> We have completed a memory analysis and will include our results in the paper. In general, we found a tradeoff between serving larger models, and serving a smaller model with CFG.
>
> The paper concludes that running a model with CFG provides comparable performance than a model with twice the parameter count. While it is clear that TRAINING a 50% smaller model is cheaper, you raise an important question about serving, and the memory impact wrt to sequence length / kv cache length. Should we serve with a model CFG or serve a vanilla model that is twice as big?
>
> Let’s call these M and M’, respectively. Let P be the parameter count of M (in floats), S the size of the kv cache, C the number of floats in the kv cache for a single token for M, and C’ the number of floats in the kv cache for a single token for M’.
> Inference on M occupies P + 2CS floats in memory
> Inference on M’ occupies 2P + C’S floats in memory
>
> The tradeoff favors CFG when VRAM is dominated by model parameters, while serving a bigger model would be optimal when the VRAM is dominated by the cache. After calculating `cache size per token = f(parameter count)`, we solve for S to find when both strategies use the same amount of VRAM, which delimits the optimal strategy change point. We end up with this chart:
>
> https://imgur.com/3nOdTqG.png
>
> For instance, if you have 10GB of VRAM, in order to maximise the VRAM available for the kv cache, models below 700M param count should be deployed without CFG. Models between 700M and 5B should be served with CFG. Models bigger than 5B can’t be served at all. (All VRAM calculations assume fp16 / bf16.)
>
> At a high level, this means that using CFG depends on your use-case. For researchers or small scale deployments where people are using the largest model that they can fit on their GPU, it's better to use CFG. However for very large scale commercial deployments, it makes more sense to increase the size of the model.
>
> We further note that increasing the size of the model is not always possible: OpenAI probably doesn't have a version of GPT-4 that's twice as big sitting around, and people using open models typically also have a maximum model size that's accessible to them. So once you wish to serve a model at least 50% as large as the maximum model size, the calculus favors CFG, regardless of VRAM.
>
> This reasoning has been added to the paper in Appendix B.3.
>
> > There is no guarantee that the Eq.6 will obtain a legal probability with the probabilities of all possibilities summing up to 1.
>
> This is a great point and we will be clearer in the paper. Our actual implementation of CFG is on the logits-level, see Equation (7). Because logits are transformed with a softmax function, we still have a valid sampling function. (Other established and popular methods alter sampling logits before softmax as well, e.g. nucleus sampling [1] and contrastive decoding [2]).
>
> Thank you again for your thoughtful comments. If you would consider raising your score in light of this response, we would really, really appreciate it!
>
> [1] Holtzman, Ari, et al. "The curious case of neural text degeneration." arXiv preprint arXiv:1904.09751 (2019).
> [2] Li, Xiang Lisa, et al. "Contrastive decoding: Open-ended text generation as optimization." arXiv preprint arXiv:2210.15097 (2022).

---

> ### Author Response · Authors · 2023-11-23
> **Thank you for your review!**
>
> Thank you for your review! We were wondering if you were planning on updating your score to reflect our reply or if there were any additional questions you'd like us to answer.

---

### Meta-Review · Area_Chair_LFD2 · 2023-12-09

**Metareview:**

In this paper, the authors explore that CFG (Classifier-Free Guidance) can be an effective way of increasing adherence to the prompt in
autoregressive language modeling. Inspired by the employment in text-to-image generation, the authors apply CFG to the logits of next-token predictions in language models. Extensive experiments show that CFG is beneficial for enhancing the performance of LLMs on NLP tasks, improving the performance of CoT and self-consistency, and increasing faithfulness and coherence.

The paper is well-written and easy to follow. However, we are concerned about the novelty and contribution of the paper as many recent works have explored CFG in language models and the authors also cannot convince the reviewers how this study differs from the other related works.

**Justification For Why Not Higher Score:**

The novelty and contribution of the paper are limited and not up to the standard of the ICLR conference

**Justification For Why Not Lower Score:**

N/A

---

### Decision · Program_Chairs · 2024-01-16

Reject